



**1  Monsoon-facilitated characteristics and transport of atmospheric**

**2  mercury at a high-altitude background site in southwestern China**

Hui Zhang[1], Xuewu Fu[1*], Che-Jen Lin[1,2], Lihai Shang[1], Yiping Zhang[3], Xinbin Feng[1*], Cynthia Lin[4]
[1] *State Key Laboratory of Environmental Geochemistry, Institute of Geochemistry, Chinese Academy of*
*Sciences, Guiyang 550002, PR China.*
[2] *Center for Advances in Water and Air Quality, Lamar University, Beaumont, Texas 77710, United States.*
[3] *Xishuangbanna Tropical Botanical Garden, Chinese Academy of Sciences, Kunming 650223, China.*
[4]*The McKetta Department of Chemical Engineering, The University of Texas at Austin, Austin, Texas 78712,*
*United States.*
*Corresponding authors: Xinbin Feng (fengxinbin@vip.skleg.cn), Xuewu Fu (fuxuewu@mail.gyig.ac.cn)*
**Abstract**
To better understand the influence of monsoonal climate and transport of atmospheric mercury (Hg) in
southwestern China, measurements of total gaseous mercury (TGM, defined as the sum of gaseous elemental
mercury, GEM, and gaseous oxidized mercury, GOM), particulate bound mercury (PBM) and GOM were carried
out at Ailaoshan Station (ALS, 2450 m a.s.l.) in southwestern China from May 2011 to May 2012. The mean
concentrations (±standard deviation) for TGM, GOM and PBM were 2.09±0.63 ng m$^{-3}$, 2.2±2.3 pg m$^{-3}$ and
31.3±28.4 pg m$^{-3}$, respectively. TGM showed a monsoonal distribution pattern with relatively higher
concentrations ($p$=0.021) during the Indian summer monsoon (ISM, from May to September) and the East Asia
summer monsoon (EASM, from May to September) periods than that in the non-ISM period. Similarly, GOM
and PBM concentrations were higher in the ISM period than in the non-ISM period. This study suggests that the
ISM and the EASM have a strong impact on long-range and transboundary transport of Hg between southwestern
China and South and Southeast Asia. Several high TGM events were accompanied by the occurrence of northern
wind during the ISM period, indicating anthropogenic Hg emissions from inland China could rapidly increase
TGM levels at ALS due to strengthening of the EASM. Most of the TGM and PBM events occurred at ALS
during the non-ISM period. Meanwhile, high CO concentrations were also observed at ALS, indicating that a
strong south tributary of westerlies could have transported Hg from South and Southeast Asia to southwestern
China during the non-ISM period. Consequently, southwestern China is an important anthropogenic source
region of ALS during the ISM period. The biomass burning in Southeast Asia and anthropogenic Hg emissions
from South Asia should be the source of atmospheric Hg in remote areas of southwestern China during the non-
ISM period.
**Keywords:** Atmospheric mercury, Indian summer monsoon, Transboundary transport, Southwestern China,
Southeast Asia



**Introduction**

Mercury (Hg), because of its volatility and long residence time in atmosphere, can transport a long distance with air mass from anthropogenic Hg emission regions to remote areas (Schroeder and Munthe, 1998;Pirrone et al., 2010). Therefore, the monsoonal climate can strongly affect the transport and distribution of atmospheric Hg in monsoon regions, such as East and South Asia. The onset of ISM in May causes air masses, originating from the Indian Ocean, to overpass South and Southeast Asia, and move northeastwardly to mainland China. Air pollutants such as $SO_2$ and CO also travel into Mainland China via air transport caused by the ISM (Xu et al., 2009;Bonasoni et al., 2010;Lin et al., 2013). In addition, the south tributary of westerlies, which passes over northern India and Myanmar into southwestern China, can also carry air pollutants to southwestern China and Tibetan plateau (Loewen et al., 2007;Xu et al., 2009;Yao et al., 2012). In East Asia, EASM is the dominant monsoon. During the monsoon period (from May to September), the warm and moist air masses from the Pacific Ocean sweep through the coastal area of China into inland China, and then move across southwestern China and the eastern Tibetan plateau. During the non-ISM period (from October to April), the dry and cold air masses from Siberia and Central Asia move through Mainland China into the Pacific Ocean via the westerlies (Hsu, 2005;Fan et al., 2013;Yu et al., 2015). The monsoonal wind changes play an important role in the transport of regional Hg emissions in Southeast and East Asia (Sheu et al., 2010a;Tseng et al., 2012;Lee et al., 2016).

An increasing number of studies have indicated that pollutant emissions and transport originate from developing countries in South and Southeast Asia (Wang et al., 2009;Lawrence and Lelieveld, 2010;Bonasoni et al., 2010;Wang et al., 2015), home to more than a billion people with strong energy demands, can pose an impact to other regions. These areas are regarded as important source regions of many air pollutants that pose significant health risk locally and regionally (Rajgopal, 2003;Lelieveld et al., 2001). Previous studies indicated that Hg emissions within South and Southeast Asia, including southwestern China, has significant impacts on the distribution and deposition of atmospheric Hg in South and East Asia (Pirrone et al., 2009;Mukherjee et al., 2009;Sheu et al., 2013;Fu et al., 2015;Zhang et al., 2012). These influences have raised concerns about/regarding high atmospheric Hg levels in India and Southwestern China, and increased Hg contents in the snow packs of Hindu Kush Himalayan-Tibetan glaciers (Loewen et al., 2005;Loewen et al., 2007;Kang et al., 2016). Previous studies reported that the open biomass burning in forests and agricultural waste burning in Southeast Asia are major sources for atmospheric Hg, aerosols and persistent organic pollutants in the region, which are subject to trans-boundary transport (Reid et al., 2013;Chang et al., 2013;Zhang et al., 2010;Sheu et al., 2013;Zhang et al., 2015;Wang et al., 2015). However, studies with respect to Hg emissions in South and Southeast Asia and the associated transboundary transport mediated by monsoonal weather are still lacking.

In this study, we conducted comprehensive measurements of TGM, GOM and PBM at Ailaoshan Station (ALS), a remote site in Southwestern China. ALS is located in the subtropical mountainous region of Yunnan province and is close to South and Southeast Asia. The air flow to ALS is mainly controlled by the Indian monsoon



climate with plenty of rainfall (85% of the total annual rainfall occurred during the ISM period) and also can be
affected by EASM during the spring through early fall. In the winter, the weather is controlled by dry and cold
monsoon circulation including westerlies and the cold Siberian current (Liu et al., 2003b;Yuhong and Yourong,
1993;ZHAO et al., 2006). Therefore, ALS is as a unique location for studying the long-range and transboundary
transport of Hg influenced by the ISM and the EAMS.

In this paper, we present the observations of TGM, GOM and PBM during the ISM and non-ISM periods at ALS,
and discuss the transboundary transport characteristics using backward trajectory analysis. We also assess the
potential contributing sources of Hg, and analyze the pathways of transboundary transport. This study is part of
the Global Mercury Observation System (GMOS, http://www.gmos.eu/), which aims to establish a global
mercury monitoring network for ambient concentrations and deposition of Hg though ground-based
observational platforms and oceanographic aircraft campaigns (Sprovieri et al., 2013)

**2 Materials and methods**
**2.1 Measurement site descriptions**
This study was conducted at Ailaoshan Mountain National Natural Reserve (24°32'N, 101°01'E) which lies in
the Yunnan province of southern China, a protected forest section covering 5100 ha on the northern crest of a
pristine evergreen broad-leaved forest on Mt. Ailao (23°35' –24°44' N, 100°54' –101°01' E). The forest altitude
ranges from 2450 to 2650 m. above sea level (a.s.l.). The climate is influenced by both ISM and EASM during
warm seasons with plenty of rainfall. On the contrary, the dry and cold monsoon circulation from the south
tributary of westerlies control the climate of Mt. Ailao in the winter. Annual mean air temperature and rainfall
in the study area are 11.3 °C and 1947 mm, respectively (You et al., 2012). Mt. Ailao is regarded as the largest
tract (504 km$^2$) of natural evergreen broad-leaved forest and one of China's most important natural areas which
has remained relatively undisturbed by human influences due to poor access (Liu et al., 2003a). Situated about
160 km to the south of Kunming, the capital of Yunnan province, ALS is relatively isolated from large
anthropogenic Hg sources. The nearest populated center is Jingdong County (Population: 36500, 1200 a.s.l.),
located 20 km to the south. Hg emissions in the Jingdong area is relatively low, ranging between 5-10 g km$^{-2}$,
as displayed in Fig.1.

**2.2 Sampling methods and analysis**
**2.2.1 Measurements of atmospheric TGM, GOM and PBM**
From May 2011 to May 2012, TGM (GEM+GOM) in ambient air was measured every 5 minutes with an
automated mercury vapor analyzer, Tekran Model 2537A (Tekran Inc., Toronto, Canada), which is widely used
for monitoring atmospheric Hg. The automated instrument collects Hg on gold cartridges and then thermally
desorbs and detects the Hg by Cold Vapor Atomic Fluorescence Spectroscopy (CVAFS). The Tekran 2537A
performs automatic calibration for TGM every 73 hours using an internal permeation source. To evaluate these





automated calibrations, manual external injections using Tekran 2505 with known concentrations of Hg were
performed every 4 months. PBM (≤0.2 μm) were removed using a 47 mm diameter Teflon filter (pore size 0.2
μm). To prevent the effect of Hg emission from ground and GOM sorption, the A Teflon sampling line with its
inlet 5 m above the ground and heat preservation (50 °C) was employed at the sampling site. To mitigate the
influence of low atmospheric pressure on the pump's strain, a low sampling rate of 0.75 L min$^{-1}$ (at standard
temperature and pressure)(Fu et al., 2008b;Swartzendruber et al., 2009;Zhang et al., 2015).

GOM and PBM was measured using a denuder-based system. The quartz denuders can collect GOM while air
passes through the KCl-coated surfaces. However, GOM and PBM have extremely low concentrations and
complex chemical reactivities in the atmosphere, and their chemical compounds are not well known. Several
previous studies reported that different GOM compounds ($HgCl_2$, $HgBr_2$ and $HgO$) have different collection
efficiencies for the KCl-coated denuder surface, as high relative humidity can passivate KCl-coated denuder and
make GOM recoveries decrease (Huang et al., 2013a;Gustin et al., 2015;Huang and Gustin, 2015). In this study,
the measurements of GOM and PBM were achieved by a manual method. The procedure of sampling and
analysis of the manual method is analogous to the Tekran speciation system using identical denuders to the
Tekran system with KCl coating (Gustin et al., 2015), differing only by manual operation. Details regarding
the measurement system and the quality assurance routines are presented in earlier works (Xiao et al.,
1997;Landis et al., 2002;Feng et al., 2000;Fu et al., 2012c;Zhang et al., 2015).

Four sampling campaigns were carried out for PBM and GOM measurements: August 17–24, 2011, December
3–17, 2011, April 12–19, 2012, and July 11–21, 2012. The selected periods represented the ISM period (May
and September) and non-ISM period (October and April) observations. Before sampling, the denuders were pre-
cleaned by pyrolysis to obtain the filed blanks, which was at 1.2 ± 0.7 pg (N=12) for denuders. The quartz fiber
filter was heated at 900 ˚C for 30 minutes for pre-cleaning. A somewhat higher field blank (6.2 ± 2.7 pg, N=20)
was observed and used to correct the PBM concentrations by subtracting the mean blank from the detected Hg.
In this study, data QA procedure followed the GMOS Standard Operation Procedure and Data Quality
Management (D'Amore et al., 2015).

**2.2 Meteorological data and backward trajectory calculation**
Meteorological parameters, including rainfall (RF), wind direction (WD), wind speed (WS), air temperature (AT)
and relative humidity (RH), were provided by the local weather station from ALS. In order to identify the
influence of long-range transport on the measured Hg at the study site, three-day backward trajectories were
calculated using HYSPLIT and the Global Data Assimilation System (GDAS) meteorological data archives of
the Air Resource Laboratory, National Oceanic and Atmospheric Administration (NOAA). The meteorological
data are of 1°×1° spatial resolutions at 6-hour intervals. All the backward trajectories ended at the sampling site



at an arrival height of 500 m above the ground. The backward trajectories were calculated at 1-hour intervals,
and cluster analysis of the trajectory endpoints was performed to determine the regional transport pathway. To
distinguish the larger sources from moderate sources, a weighing algorithm based on measured concentrations
(concentration weighted trajectory (CWT)) was applied in this study. In this procedure, each grid cell received
a source strength obtained by averaging sample concentrations that have associated trajectories that crossed that
grid cell as follows:
$$C_{ij} = \frac{1}{\sum_{l=1}^{M} \tau_{ijl}} \sum_{l=1}^{M} C_l \tau_{ijl}$$

$C_{ij}$ is the average weighted concentration in the grid cell $(i,j)$. $C_l$ is the measured Hg concentration, $\tau_{ijl}$ is the
number of trajectory endpoints in the grid cell $(i,j)$ associated with the $C_l$ sample, and $M$ is the number of samples
that have trajectory endpoints in grid cell $(i,j)$. A point filter is applied as the final step of CWT to eliminate grid
cells with few endpoints. Weighted concentration fields show concentration gradients across potential sources.
This method helps determine the relative significance of potential sources (Hsu et al., 2003;Cheng et al., 2013).
**3 Results and discussion**
**3.1. Features of monsoonal transport and characteristics observed Hg species**
**3.1.1 General distribution characteristics of TGM in atmosphere**
The highly time-resolved long-term data set of TGM concentrations in ambient air at ALS is displayed in Fig.
2, and the mean TGM concentration over the sampling period was 2.09±0.63 ng m⁻³ with a higher level (2.22 ng
m⁻³)during the ISM period than that during the non-ISM period (1.99 ng m⁻³) (Table 1). The TGM mean
concentration at ALS was slightly higher than that of the global background (1.5-1.7 ng m⁻³ in the Northern
Hemisphere and 1.1-1.3 ng m⁻³ in the Southern Hemisphere (Lindberg et al., 2007;Slemr et al., 2015;Venter et
al., 2015), and higher than those (1.58 to1.93 ng m⁻³) observed in some remote areas in northern America and
Europe (Kim et al., 2005;Sprovieri et al., 2010). Compared to the background concentrations observed at the
Shangri-La Baseline Observatory in Yunnan province (2.55±0.73 ng m⁻³, (Zhang et al., 2015), at Mt. Leigong in
Guizhou province (2.80 ± 1.51 ng m⁻³, (Fu et al., 2010b) and at Mt. Gongga in Sichuan province 3.98±1.62 ng
m⁻³, (Fu et al., 2008a), the mean TGM level at ALS was lower. However, the mean TGM level at ALS was higher
than those observed at Mt. Changbai (1.60 ± 0.51 ng m⁻³) in Northeast China and at Mt. Waliguan (WLG)
Baseline Observatory (1.98 ± 0.98 ng m⁻³) in the Tibetan plateau (Fu et al., 2012a;Fu et al., 2012b). Interestingly,
most peaks of high TGM concentrations at ALS frequently appeared during the ISM period (Fig. 2). This differed
from the previous results at Mt. Gongga and Mt. Leigong of southwestern China but was similar to the results
at Shangri-La. There were also several peaks that appeared during the non-ISM period, which could have been
caused by different sources than those during the    ISM period. The sampling site is located adjacently to South
Asia and Southeast Asia, and Hg emissions from biomass burning in South Asia and Southeast Asia would
inevitably contribute to the elevated TGM concentrations at ALS during the non-ISM period (Wang et al., 2015).
Southwestern China is one of the largest Hg emission areas in China, and coal combustion and non-ferrous metal





(especially zinc) smelting activities are the two main Hg sources. It was reported that total Hg emission from
Guizhou, Sichuan and Yunan provinces reached about 128 tons in 2003 (Wu et al., 2006), and the large amount
of Hg emissions contributed to the elevation of TGM concentrations in this area. Since Guizhou, Sichuan and
Yunan provinces are located in the upper wind direction of the sampling site to EASM, Hg emission from these
areas can be transported to ALS and result in the elevation of TGM concentrations.

### 3.1.2 Monthly TGM anomalies and wind meteorology

To assess the monsoonal variation of TGM concentrations, the distribution of monthly mean TGM
concentrations at ALS is shown in Fig. 3. The Hg concentrations during the ISM period were higher than those
during the non-ISM period. The highest monthly concentration was observed in May with a mean value of 2.46
ng m$^{-3}$, and the lowest monthly mean concentration of 1.45 ng m$^{-3}$ was observed in November. Although there
were relatively higher Hg levels in December and January during the non-ISM period, this pattern was generally
different from the most common pattern in the Northern Hemispheric which has a summer minimum and winter
maximum TGM distribution pattern as observed in many previous studies (Kellerhals et al., 2003;Kock et al.,
2005;Fu et al., 2010a). There were several possible reasons for this monsoonal distribution pattern of TGM
concentrations on ALS.

Firstly, the increase of TGM concentrations during the ISM period could be due to the interaction of the EASM
and the ISM, promoting the air masses with high Hg from the areas of anthropogenic Hg emissions to ALS.
Generally, ALS is located on the low latitude highlands of Yunnan in southwestern China which is subject to
the interactions between the EASM and the ISM, although most of time, the air flow of Yunnan is mainly
controlled by the ISM during the ISM period. However the strengthening of the EASM or the weakening of the
ISM can also spur the EASM to control this area and bring precipitation during the ISM period (Fan et al., 2013).
Therefore, the TGM level should be sensitive to the strengthening/weakening of the two monsoons. Once the air
flow from high Hg source regions (Sichuan, Guizhou and Chongqing) is transported to ALS with the
strengthening of the EASM, TGM levels at ALS can increase rapidly. However, anthropogenic Hg emissions
from inland China could increase the Hg background level with the raid of westerlies and cold Siberian current
during the non-ISM period. Previous studies discussed the seasonal change of TGM at the background sites of
southwestern China and found that increased domestic coal consumption and an increase in household heating
was the main cause of elevated TGM concentrations observed in winter (Fu et al., 2008a;Fu et al., 2010a).
Additionally, the biomass burning in Southeast Asia could also be an important reason for high Hg level at ALS
during the non-ISM period. Intense biomass burning originating from Southeast Asia typically occurred in late
winter and spring (Huang et al., 2013b). This could be the cause of the high TGM at ALS along with the long-
range transboundary transport in the spring (Wang et al., 2015).

Fig. 4 displays the distribution frequency of TGM above and under the average (2.09 ng m$^{-3}$) based on wind



direction including Northeast (NE), Southeast (SE), Southwest (SW) during the ISM and non-ISM period. It is
clear that SW was the predominating wind direction, and there was no Northwest (NW) during the entire study
period. The SW frequency was highest when high and low Hg levels occurred during the ISM period or non-
ISM period, and the SW frequency showing low Hg was higher than that of high Hg. This could be the reason
why the average Hg level from SW was not high. The air flows originating from South Asia and Southeast Asia
could contribute to high Hg concentrations at ALS. Contrarily, NE and SE frequency had a relatively lower trend
than SW, but high Hg frequency from NE and SE were both high during the ISM period. This should be the
result from the strengthening of EASM during the ISM period. However, during the non-ISM period, the cold
and dry air flow from the south tributary of westerlies could have swept over South Asia and Southeast Asia and
moved to ALS with high wind speed (Fig. 3). This dry air flow could have also taken the air masses of high Hg
levels emitted from biomass burning in South Asia and Southeast Asia to ALS and caused a rapid increase of
Hg level at ALS. In addition, cold air flows could also transport Hg emitted from inland China to ALS due to
the strengthening of the cold Siberian current during the non-ISM period. Therefore, there were some high TGM
events in December and March at ALS (Fig. 3).

**3.1.3 Seasonal variation of GOM and PBM influenced by monsoonal weather**
Table 1 shows seasonal statistics of daily averages for Hg species and select meteorological parameters which
were determined on a seasonal basis and for the year-long dataset. TGM during the ISM period was statistically
higher than during the non-ISM period (Table S1). Meanwhile, AT, RF and RH had a monsoonal distribution
with the highest level during the ISM period, and SW frequency had decline with increase of SE and NE
frequency during the ISM period. This suggests the EASM could also influence the climate at ALS during the
ISM period, which was consistent with TGM concentration that the site is impacted by regional sources including
biomass burning and monsoonal long-range transboundary transport.

For GOM and PBM, which on average accounted for <2% of the TGM, there were also seasonal trends. Both
species had the highest levels in autumn while GOM was lowest in the winter and PBM was lowest in the spring.
The lowest GOM level was observed in the summer, which increased consistently to reach the highest value
($3.4\pm3$ pg m$^{-3}$) in autumn. Similarly, PBM displayed the same distribution in seasonal variation, with the highest
PBM level in autumn ($46.3\pm28.8$ pg m$^{-3}$). Meanwhile, the AT, RF and RH were also higher during the ISM
period than those during the non-ISM period. However, unlike TGM, the GOM and PBM were closely linked
with atmospheric Hg chemistry, meteorological patterns, and numerous other factors. Thus, there are several
likely factors that contribute to these trends, including a greater number of sources during the ISM period, and
changing ecological or meteorological conditions. Previous studies in the Mt. Gongga and Mt. Leigong area
suggested that enhanced coal and biomass burning played a significant role in elevated TGM concentrations
during cold seasons (Fu et al., 2008a;Fu et al., 2010a). Enhanced coal and biomass burning during cold seasons
is generally driven by the need for residential heating in China. However, in the southern Yunnan province, the





air temperature is high during the non-ISM period. Thus, the domestic use of coal is not dominant for residential
heating, but rather the agricultural activity in the region including the crop harvesting and the burning of straw.
This was probably one of most important reasons for the highly elevated GOM and PBM level at ALS in autumn.

Monsoonal distribution patterns and mean TGM, GOM and PBM concentrations based on the four sampling
campaigns in ALS are shown in Fig. 5. Mean concentration of the three Hg species showed a monsoonal
variation with higher levels during the ISM period than during the non-ISM period. This suggests that regional
anthropogenic emissions are important Hg sources in southwestern China. During the ISM period, not only was
air flow originating from the Indian Ocean dominating, but air flow that occasionally originated from the Pacific
Ocean also intruded the study site, which passed through central and southwestern China, one of the most Hg-
polluted regions. Moreover, the TGM, GOM and PBM levels from the north wind were higher than those from
the south wind to ALS (Fig. S1). Thus, the air masses likely captured large amounts of Hg during transport and
caused elevated atmospheric Hg concentrations at ALS during the ISM period.

Fig. 6 shows pollution roses of TGM, GOM and PBM at ALS during the ISM period and during the non-ISM
period respectively. The wind direction at the study site was dominantly SW. This reflects that the predominant
monsoon influence the ALS site is the ISM and westerlies. During the ISM period, most of the TGM, GOM and
PBM events were from SW, slightly fewer were from NE, and both SW and NE exhibited higher TGM, GOM
and PBM events. This indicates SW and NE were the two primary directions of high Hg sources during the ISM
period. However, during the non-ISM period, almost all TGM, GOM and PBM events were from SW. This
indicates that strong westerlies were the primary winds during the non-ISM period. These westerlies could take
the Hg from South Asia and Southeast Asia into southwestern China. Thus, the dependence of atmospheric Hg
species on wind was likely attributed to an interplay of regional sources and the long-range transboundary
transport of Hg.

Indeed, GOM concentrations were extremely low at ALS. Marine air masses during the ISM period likely diluted
the atmospheric Hg in the study area. The high RF and low WS can promote the wet deposition of GOM and
PBM. Therefore, in the summer, RH was very high, but the GOM and PBM were very low at ALS. A new study
reported that high RH could reduce the collection of GOM by the KCl-coated denuder (Huang, Gustin et al.
2015). This could be another reason why the GOM was low in summer. Additionally, low GOM and PBM could
be also related to rapid deposition of Hg due to the high altitude montane environment and luxuriant virgin forest
cover of Mt. Ailao. A previous study already found that the increasing occurrence and extension of fog and cloud
droplet interception can enhance the uptake of Hg by foliage (Zhang et al., 2013). We will study the possible
reasons why the GOM level is exceedingly low in ALS via continuous long-term monitoring for GOM in the
future.



**3.2 Transboundary transport of Hg facilitated by monsoons**
The backward trajectories arriving at ALS over the study period were grouped into five clusters, which are shown
in Fig. 7. Most of these backward trajectories consisted of air masses that originated from the South Asia and
Southeast Asia, passing over India, Bengal Bay, the Indo-China peninsula and the southern Yunnan province of
China. Just 4.7% of air masses originated from inland China and then passed over Sichuan, Guizhou, Yunnan
provinces and the city of Chongqing, China. For the five types of air masses, air masses in cluster 1 displayed
very low TGM concentrations (1.86 ng m$^{-3}$) for all the air masses, although their frequencies were the highest
(38.55%). Similarly, the mean TGM concentration in cluster 3 was 1.92 ng m$^{-3}$, which was also considerably
low and had second highest frequency (23.03%). This suggest that Hg emitted in South Asia could not have
largely contributed to the high Hg levels at ALS. Air masses in cluster 4 showed a high TGM concentration of
2.42 ng m$^{-3}$, which originated over the South China Sea and passed over northern Vietnam and Laos and the
southern Yunnan and Guangxi provinces of China, which are generally areas of less anthropogenic emissions
other than biomass burning during the non-ISM period. However, the air masses in cluster 5 were also polluted
with Hg, with a mean concentration of 2.20 ng m$^{-3}$, which is higher than those of clusters 1, 3 and 4. Air flows
likely originated in Bengal Bay and passed over Myanmar since most anthropogenic emissions in Myanmar are
centralized in southern Myanmar, and intense biomass burning in the area during the non-ISM period perhaps
contributed to a slightly high TGM level of these air masses. Cluster 2 displayed the highest TGM concentrations
(2.65 ng m$^{-3}$). Air masses in cluster 2 passed over the inland China region, which is the most densely populated
and heavily Hg-polluted area in China due to industrial and domestic coal combustion, smelting industries,
cement production, biomass burning, etc. Sichuan, Guizhou, Chongqing and the northwestern Yunnan provinces,
respectively, were contributing Hg source provinces in China. Although only 4.70% of air masses were from the
southwestern China region, the TGM concentrations of these air masses were the highest, which could be an
important reason for elevated TGM levels at ALS.

The Tibetian plateau and Yunnan-Guizhou plateau are located in north of ALS, which is a monitoring site at
high elevation. To the south of ALS, the geography consists of a montane area plain of low elevation. Thus, the
air masses at ALS were from different directions and of different heights, which may have affected the Hg level
in the air masses as they passed over different anthropogenic emission regions. Fig. S2 compares the three-
dimensional height of all the wind clusters arriving at ALS. The height of cluster 1 from India was the highest.
Such transport pattern tends to more effectively dilute Hg emissions from low altitude surface to ALS. Thus, in
non-ISM period, the south tributary of westerlies passed over India does not lead to elevated Hg concentrations
at ALS. Additionally, due to the high Hg concentration in the air in southwestern China, the TGM level of any
air masses coming from the northeast of ALS should be increased at ALS regardless of height. Contrarily, cluster
3 had a high height but low level TGM because its air masses originated and passed over the area of low
anthropogenic emission region.




### 3.3 Impacts of Hg emission from industrial sources and biomass burning

Fig. 8 shows the seasonal trend analysis using the average of the daily TGM values, RF distribution, WS and WD during the study period along with IMI (Indian monsoon index) at ALS. In general, May to September represent the normal ISM period. The IMI, however, also illustrates the onset in of the ISM in May and its retreat at the end of September. During the ISM period, the mean TGM is 2.22 ng m$^{-3}$, which is slightly higher than 1.99 ng m$^{-3}$ during the non-ISM period, and most of the RF events appeared during the ISM period. More crucially, some high-frequency NE and east wind events occurred during the ISM period. Once the WD shifted from SW to NE, the TGM level rapidly increased in addition to relatively lower RF and WS. This indicates that the strengthening of EASM can move the air masses with high Hg levels from inland China to ALS, but when the NE air flow climbed over the Yunnan-Guizhou plateau, the speed of air mass movement decreased and carried less RF. However, during the non-ISM period, the high TGM events did not accompany the appearance of NE. The WD was primarily SW other than a few SE at the end of March, and the WS was higher than the level during the ISM period. This indicates that the south tributary of westerlies could control the climate and carry air masses from South Asia and Southeast Asia. High Hg events were also evident with SW and SE during the non-ISM period, which could be due to Hg emission from biomass burning in South Asia and Southeast Asia.

Therefore, five special high TGM events were accompanied by an exceeding variation of WD and WS to analyze the reasons of high TGM appearance. As Fig. 8 shows, five extreme peaks of high TGM were displayed from June 23-29 and July 10-18, 2011, September 28 to October 9 and December 23-31, 2011, and March 23-29, 2012, respectively. On June 23, the air masses from India began to increase TGM concentrations, and with the strengthening of the EASM, the TGM level increased gradually and peaked with low RF and WS on June 26. Meanwhile, the air masses that had just swept Chongqing, Sichuan, Guizhou and eastern Yunnan provinces, including Kunming city, where there is high Hg emission because of industrial activities (Fig. 1). Thus the air flow from these areas could suddenly increase the TGM level at ALS due to the long-range transport. Once the air flow from EASM faded away, the ISM would control the ALS area again, the air flow would shift to southwest, and the TGM level would return to average levels (Fig. 9a). The same variation appeared from July 10-18 during the ISM period. TGM levels increased on July 10 and were highest (3.65 ng m$^{-3}$) from July 14-16 with the strengthening of EASM. When WD shifted to SW, TGM was back to its base level (Fig. 9b). In fact, during the ISM period, this sort of peak appeared many times (Fig. 2), the reason for these peaks being similar to the two peaks in June and July.

However, on September 28, due to the fadeaway of ISM and the strengthening and incursion of air flow from EASM as well as the cold Siberian current, a high TGM event was initiated as the air flow shifted from SW to NE (Fig. 10). When the air flow swept Chongqing, Sichuan, Guizhou and eastern Yunnan province again from September 30 to October 5, the TGM level increased to its highest level (3.18 ng m$^{-3}$), then gradually decreased with the shift in air flow that swept northern Vietnam and Laos. These high Hg events happened during the



transitional period from the ISM period to the non-ISM period, which indicates that the high Hg emission from
inland China had severely influenced the TGM level at ALS. As previously discussed, the strengthening of the
EASM or the weakening of ISM caused the air flow with high Hg originating from inland China to be transported
to ALS, which contributed to extremely high TGM concentrations.

During the non-ISM period, with the fading of the ISM and EASM, the south tributary of westerlies grew
stronger as the dry and cold air flow swept over South and Southeast Asia and arrived in southwestern China to
control the climate. Hence, industrial sources likely contributed to the high level Hg concentrations in this study
site. In Fig. 11, when air masses entered from important Southeast Asia industrial regions (e.g. Hanoi, Haiphong
et al.), the TGM level was highest (3.13 ng m$^{-3}$) while TGM concentrations decreased 11% with the entrance of
the strong wind from the Bay of Bengal. In addition, fire events were also observed along the backward
trajectories. Moreover, the high correlation ($R^2$=0.89) between Hg and CO was identified in Fig. 12. However,
the TGM/CO ratio was 0.01124 ng g$^{-1}$ ppb$^{-1}$ (1.80 E-6 mol mol$^{-1}$), which is more than 10 times higher than the
reported world average biomass burning ratio but was close to ratios observed in Taiwan in October (1.28 E-6,
verified from anthropogenic plumes) Sheu et al. (2010b);(Friedli et al., 2009). Therefore, in this event, the major
contributor should be the industrial sources from Southeast Asia. Fig. 13 shows that biomass burning is an
important source for Hg. On March 23, the high TGM concentrations began to arise because of the fire events
that occurred in northeastern India and north Myanmar. TGM levels were highest (4.53 ng m$^{-3}$) when the air
flow shifted and swept inland China. When the air flow shifted from east to southwest and swept south Myanmar
from March 25-28, a period of were high-frequency fire events, the TGM concentrations maintained a high level
(3.11 ng m$^{-3}$). However, when the air flow shifted to northern Myanmar, the TGM level returned to low levels
(2.01 ng m$^{-3}$), indicating that biomass burning originating from Southeast Asia could also rapidly input and
increase the TGM levels in southwestern China during the non-ISM period. Different from Fig. 12, the TGM/CO
ratio decreased by half in Fig. 14 and was comparable with reported biomass burning ratios (3.00 E-7) in Canada
and the USA (Sigler et al., 2003). Moreover, the close correlations between TFRP (total fire radiative power),
which provides information on the measured radiant heat output of detected fires, and CO ($R^2$=0.98) and TGM
($R^2$=0.45) verify the above hypothesis (Wooster et al., 2005).

**3.4 Potential source regions of atmospheric Hg**
Fig. 15a shows the possible source regions and pathways of atmospheric Hg at ALS during the ISM period
identified by the CWT analysis. Sichuan, Guizhou, Chongqing, Yunnan and Guangxi provinces in southwestern
China as well as in Southeast Asia as well as northern Laos, Cambodia, Thailand and Vietnam were likely source
regions of high atmospheric Hg at ALS during the ISM period. Southwestern China, including Sichuan,
Chongqing, Guizhou, and Yunnan provinces, is an important anthropogenic source region of China (Wang et al.,
2006;Feng and Qiu, 2008;Wu et al., 2006;Jiang et al., 2006). In fact, several capital cities including Kunming,
Guiyang and Chongqing are located about 200 to 800 km north of ALS. These capital cities may be the source



of much of the Hg emissions during atmospheric transport. The identified source areas correspond very well
with the anthropogenic Hg emission inventories in East and South Asia. The potential area identified in Southeast
Asia is also classified as a high anthropogenic Hg emission region by Hg emission inventories (Pacyna et al.,
2010;Pirrone et al., 2010;Li et al., 2009), and the biomass burning in Southeast Asia could cause high Hg
emissions. During the non-ISM period, as displayed in Fig. 15b, Northern India is an important urbanized and
industrialized area that may produce high anthropogenic Hg emission rates, as there are a number of large scale
industries and coal-fired power plants in India (Burger Chakraborty et al., 2013;Pervez et al., 2010). Additionally,
the biomass burning in Southeast Asia should be a large contributor of high TGM levels at ALS (Wang et al.,
2015). A small region in eastern Myanmar and northern Laos and Thailand was also identified as a potential
source region and pathway for TGM at ALS. The high CWT values in this area may be primarily due to high
Hg emission rates because of biomass burning during the non-ISM period.

Indeed, the emission of Hg from biomass burning includes forest fire and agricultural waste burning in Southeast
Asia and southwestern China and could also play an important role in TGM distribution and transboundary
transport at ALS. Southeast Asia and southwestern China are large tropical rain forest areas. During the ISM
period, the air had a very high RH and the highest RF. Because the fire events in Southeast Asia and southwestern
China were not frequent and the biomass burning for agriculture was not prevalent, less Hg from biomass burning
was released into the atmosphere. As shown in Fig. S3a, fire events in Southeast Asia and southwestern China
during the ISM period (mainly from June to September 2011) exhibited a much lower frequency, indicating that
the significant impact of Hg transport was primarily from anthropogenic Hg emissions. Nevertheless, during the
non-ISM period (mainly from December 2011 to April 2012), high-frequency fire events in Southeast Asia and
southwestern China were observed (Fig. S3b). Hg emissions from fire events peaked when the most intense
biomass burning occurred in Southeast Asia and Southwest China. This increased TGM level at ALS once air
masses from these areas were transported to ALS.

**4 Conclusions**
This study made at ALS suggests a significant impact of monsoonal climates on the distribution and long-range
transport of atmospheric Hg in southwestern China and shows a pronounced monsoonal variation with a high
TGM level during the ISM period and a low TGM level during the non-ISM period. This seasonal variation
opposes the previous distribution of atmospheric Hg observed in the background sites of southwestern China.
This behavior seems to be dominated by the seasonal variation of monsoons and the influence of the long-range
and transboundary transport of Hg from high anthropogenic Hg emissions and biomass burning. The high Hg
regional sources from inland China were an important factor in the elevated TGM level. Meanwhile, with the
economy developing rapidly, anthropogenic Hg emission in Southeast Asia is also increasing. This will elevate
the TGM level in the area, including that of southwestern China. Additionally, Hg emitted from India can travel
to southwestern China due to the ISM in the summer and a strong south tributary of westerlies during cold



seasons. Biomass burning includes high-frequency fire events in Southeast Asia that could play an important
role for high TGM levels during the non-ISM period. Thus, the TGM concentrations during different seasons in
addition to the impacts of monsoonal climate may pose an important constraint on the global models of
atmospheric Hg.

**Acknowledgments.** This work is supported by the National "973" Program of China (2013CB430003), the
National Science Foundation of China (41430754, 41473025, 41273145,41173024). We also thank Xin Luo,
Ben Yu and Jun Zhou for field sampling assistance.

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





**Figure Captions**

**Table 1:** The statistics for mercury species and meteorological variables based on daily averages from May
2010 through May 2011 at the ALS site.
**Fig. 1:** Map showing the location of ALS, anthropogenic mercury emissions (g km$^{-2}$ y$^{-1}$) and major cities in Asia
(AMAP/UNEP, 2013).
**Fig. 2:** Total gaseous mercury (TGM) concentration in the ambient air at ALS. Source 1 represents the peaks of
high TGM concentrations at ALS during the ISM period, which were caused by anthropogenic Hg emissions
from inland China due to the strengthening of EASM. Source 2 represents the peaks of high TGM concentrations
at ALS during the non-ISM period, which were caused by the biomass burning from Southeast Asia and
anthropogenic Hg emissions from South Asia.
**Fig. 3:** Monthly TGM anomalies at ALS. The Hg concentration during the ISM period (May to September) was
higher than that of the non-ISM period (October to April). The highest monthly concentration (2.46 ng m$^{-3}$) was
observed in May, and the lowest monthly mean concentration (1.45 ng m$^{-3}$) was observed in November. The air
temperature was higher, and the wind speed was lower during the ISM period than that of the non-ISM period.
**Fig. 4:** Distribution frequency of TGM above and under the average (2.09 ng m$^{-3}$) based on wind direction
during the ISM and the non-ISM period. The high and low Hg levels of the SW frequency was higher than
both the high and low Hg levels of the NE and SE frequencies.
**Fig. 5:** TGM, GOM and PBM variation during the ISM and non-ISM period based on the four sampling
campaigns. TGM, GOM and PBM concentrations showed a monsoonal variation with higher level during the
ISM period than in the non-ISM period.
**Fig. 6:** Pollution roses of Hg species. Most of the TGM, GOM and PBM were from SW, and some higher TGM,
GOM and PBM events were from NE during the ISM period. Almost all TGM, GOM and PBM were from SW
during the non-ISM period.
**Fig. 7:** Air mass backward trajectory analysis for long range transport to ALS. Most of these backward
trajectories consisted of air masses that originated from South and Southeast Asia and were accompanied by
lower Hg levels. Air masses originating from inland China were less frequent and accompanied by higher Hg
levels.
**Fig. 8:** The distribution of the Indian monsoon index (IMI), TGM and rainfall at ALS. IMI displays that the ISM
period was from May to September. RF events mainly appeared during the ISM period. Some high Hg events
were accompanied by NE and east wind events during the ISM period. High TGM events were accompanied by
the appearance of SW during the non-ISM period.
**Fig. 9:** Backward trajectories of air masses and the variation of TGM concentrations because of the strengthening
of and incursion of air flow from EASM on June 23-28, 2011 (a), and July 10-18, 2011 (b), during the ISM
period (May to September). The TGM level (C) were 3.27 ng m$^{-3}$(a) and 3.65 ng m$^{-3}$(b) with the air masses from





important industrial regions of inland China. The TGM level were down to 2.49 ng m$^{-3}$(a) and 3.02 ng m$^{-3}$(b)
while the air flow shifted to southwest and swept Southeast Asia.
**Fig. 10:** Backward trajectories of air masses and the variation of TGM concentrations because of the
strengthening and incursion of air flow from EASM and the cold Siberian current from September 28 to October
9, 2011, during the non-ISM period (October to April). The TGM level (C) were 2.25 ng m$^{-3}$, 3.18 ng m$^{-3}$ and
2.53 ng m$^{-3}$ with the air masses from inland China. The TGM level was down to 1.89 ng m$^{-3}$ with the air masses
from South China Sea, northern Vietnam and Laos.
**Fig. 11:** Backward trajectories of air masses and the sites of fire events from December 23-31, 2011, during the
non-ISM period (October to April). The TGM level (C) was 3.13 ng m$^{-3}$ with air masses from important
Southeast Asia industrial regions. The TGM level dropped to 2.78 ng m$^{-3}$ with air masses from the Bay of Bengal.
**Fig. 12:** Correlation of TGM and CO in December 23-31, 2011, during the non-ISM period.
**Fig. 13:** Backward trajectories of air masses and the sites of fire events in March 23-29, 2012, during the non-
ISM period (October to April). The TGM level (C) were 4.53 ng m$^{-3}$ with the air masses from inland China. The
TGM level was 3.11 ng m$^{-3}$ with the air masses from Myanmar and high-frequency fire events.
**Fig. 14:** Correlation of TGM and CO ($R^2$=0.53), correlations between TFRP (total fire radiative power) and CO
($R^2$=0.98) and TGM ($R^2$=0.45) in March 23-29, 2012, during the non-ISM period (October to April).
**Fig. 15:** Potential source regions and pathways of atmospheric TGM at ALS as identified by the CWT during
the ISM period (a, May to September) and the non-ISM period (b, October to April).



**Table 1:** The statistics for mercury species and meteorological variables based on daily averages from May
2010 through May 2011 at the ALS site.

| | | Spring | Summer | Autumn | Winter | ISM period | Non-ISM period | Total |
|---|---|---|---|---|---|---|---|---|
| TGM (ng m$^{-3}$) | Mean | 2.18 | 2.20 | 1.92 | 2.04 | 2.22 | 1.99 | 2.09 |
| | ±SD | 0.67 | 0.60 | 0.64 | 0.58 | 0.58 | 0.66 | 0.63 |
| | Range | 1.01-5.70 | 1.15-3.79 | 1.11-3.59 | 0.99-4.45 | 1.01-3.79 | 0.99-5.70 | 0.99-5.70 |
| GOM (pg m$^{-3}$) | Mean | 2.31 | 2.04 | 3.42 | 1.83 | 2.45 | 2.06 | 2.22 |
| | ±St.Dev | 1.79 | 1.39 | 2.99 | 2.85 | 2.08 | 2.40 | 2.28 |
| | Range | 0.13-10.20 | 0.27-6.68 | 0.11-13.29 | 0.12-17.25 | 0.11-13.29 | 0.12-17.25 | 0.11-17.25 |
| PBM (pg m$^{-3}$) | Mean | 22.39 | 32.19 | 46.28 | 31.97 | 36.38 | 27.36 | 31.27 |
| | ±St.Dev | 19.05 | 30.56 | 28.80 | 30.63 | 30.62 | 26.09 | 28.44 |
| | Range | 0.87-135.79 | 5.84-165.01 | 16.16-120.99 | 3.84-139.65 | 5.84-165.01 | 0.87-139.65 | 0.87-165.01 |
| AT (°C) | Mean | 13.36 | 15.53 | 11.42 | 6.96 | 15.45 | 8.93 | 11.95 |
| | ±St.Dev | 3.03 | 1.19 | 3.39 | 1.88 | 1.29 | 2.93 | 3.99 |
| | Range | 4.88-18.38 | 11.66-17.74 | 4.78-16.18 | 1.65-10.83 | 11.32-18.38 | 1.65-15.16 | 1.65-18.38 |
| RH (%) | Mean | 72.26 | 90.77 | 91.11 | 73.58 | 87.43 | 75.77 | 81.17 |
| | ±St.Dev | 15.06 | 5.56 | 4.94 | 18.17 | 9.58 | 17.48 | 15.49 |
| | Range | 38.42-97.63 | 71.17-99.25 | 79.96-100.00 | 33.17-100.00 | 52.25-99.25 | 33.17-100.00 | 33.17-100.00 |
| WS (m/s) | Mean | 3.60 | 2.59 | 2.71 | 4.29 | 2.59 | 3.80 | 3.29 |
| | ±St.Dev | 0.42 | 0.47 | 0.38 | 0.07 | 0.45 | 0.55 | 0.79 |
| | Range | 2.99-3.95 | 2.05-3.19 | 2.17-2.97 | 4.20-4.37 | 2.05-3.19 | 2.97-4.37 | 2.05-4.37 |
| RF (mm) | Total | 334.3 | 845.4 | 612.4 | 44.2 | 1493 | 343.3 | 1836.3 |
| WD (%) | NE | 3.25 | 15.22 | 14.03 | 0 | 11.35 | 5.46 | 7.79 |
| | SE | 9.76 | 14.13 | 20.65 | 4.0 | 16.22 | 10.33 | 13.07 |
| | SW | 86.99 | 66.30 | 64.13 | 95.60 | 71.89 | 84.98 | 78.89 |
| | NW | 0 | 0 | 0 | 0 | 0 | 0 | 0 |



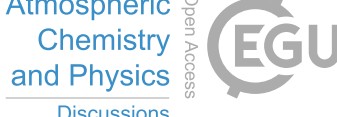



**Fig. 1:** Map showing the location of ALS, anthropogenic mercury emissions (g km$^{-2}$ y$^{-1}$) and major cities in Asia
(AMAP/UNEP, 2013).

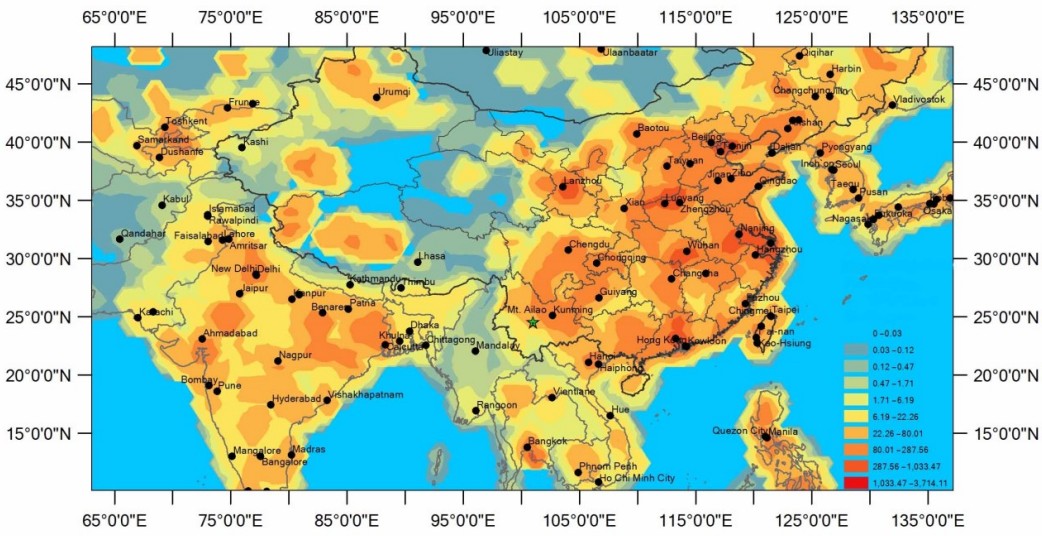






**Fig. 2:** Total gaseous mercury (TGM) concentration in the ambient air at ALS. Source 1 represents the peaks of
high TGM concentrations at ALS during the ISM period (May to September), which were caused by
anthropogenic Hg emissions from inland China due to the strengthening of EASM. Source 2 represents the peaks
of high TGM concentrations at ALS during the non-ISM period (October to April), which were caused by the
biomass burning from Southeast Asia and anthropogenic Hg emissions from South Asia.

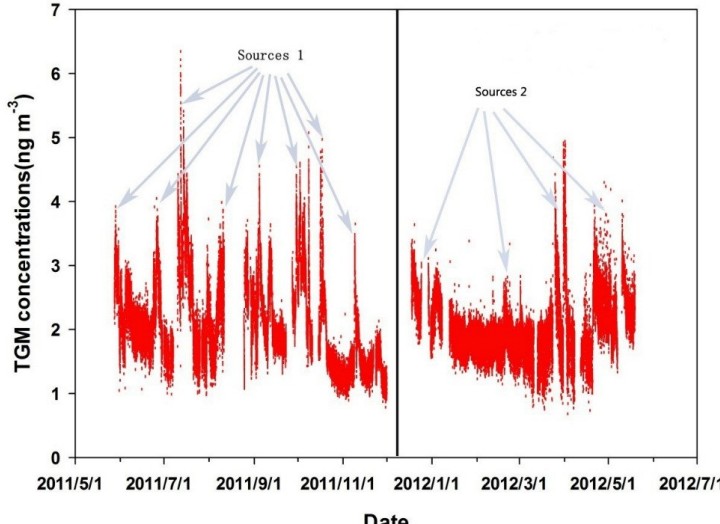






**Fig. 3:** Monthly TGM anomalies at ALS. The Hg concentration during the ISM period (May to September) was
higher than that of the non-ISM period (October to April). The highest monthly concentration (2.46 ng m$^{-3}$) was
observed in May, and the lowest monthly mean concentration (1.45 ng m$^{-3}$) was observed in November. The air
temperature was higher, and the wind speed was lower during the ISM period than that of the non-ISM period.

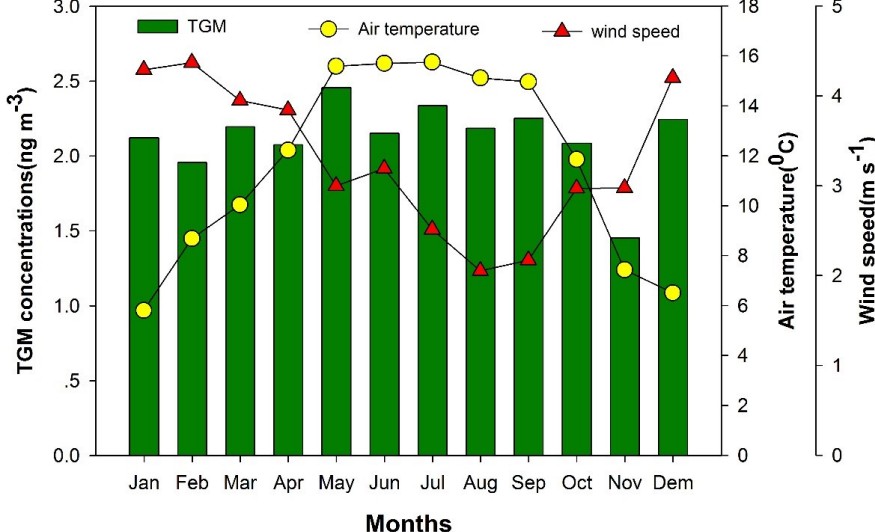






**Fig. 4:** Distribution frequency of TGM above and under the average (2.09 ng m$^{-3}$) based on wind direction
during the ISM (May to September) and the non-ISM period (October to April). The high and low Hg levels of
the SW frequency was higher than both the high and low Hg levels of the NE and SE frequencies.

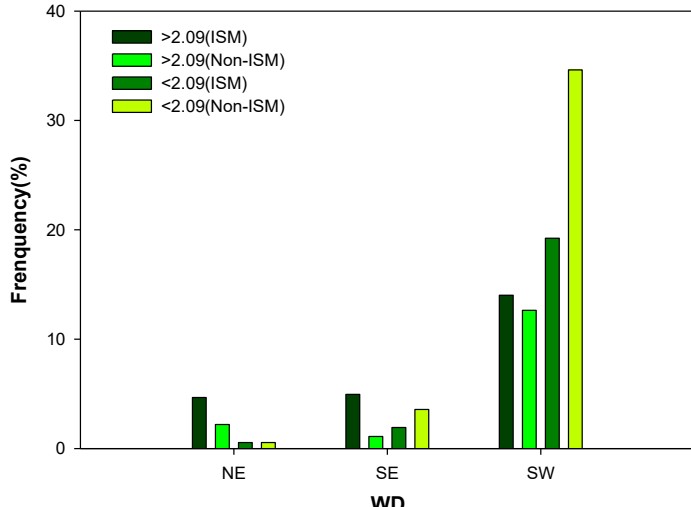





**Fig. 5:** TGM, GOM and PBM variation during the ISM (May to September) and non-ISM period (October to
April) based on the four sampling campaigns. TGM, GOM and PBM concentrations showed a monsoonal
variation with higher level during the ISM period than in the non-ISM period.

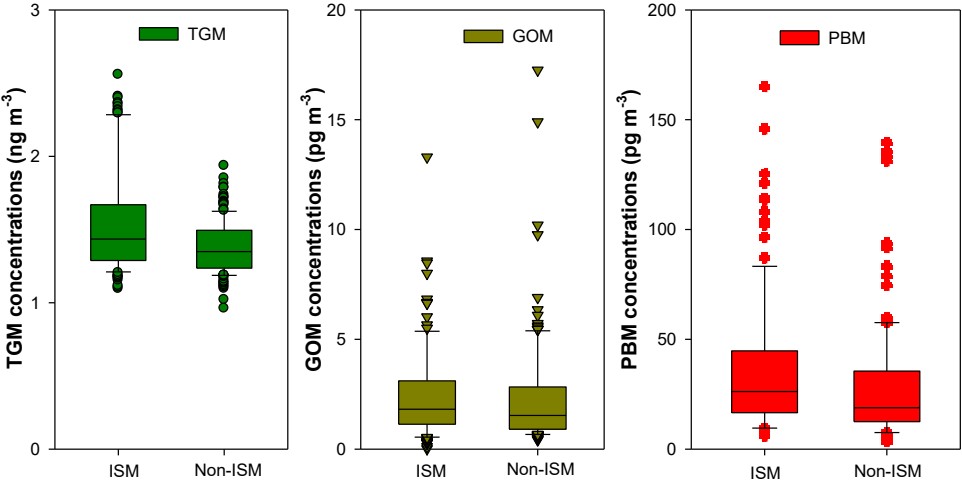






**Fig. 6:** Pollution roses of Hg species. Most of the TGM, GOM and PBM were from SW, and some higher TGM,
GOM and PBM events were from NE during the ISM period (May to September). Almost all TGM, GOM and
PBM were from SW during the non-ISM period (October to April).





**Fig. 7:** Air mass backward trajectory analysis for long range transport to ALS. Most of these backward
trajectories consisted of air masses that originated from South and Southeast Asia and were accompanied by
lower Hg levels. Air masses originating from inland China were less frequent and accompanied by higher Hg
levels.

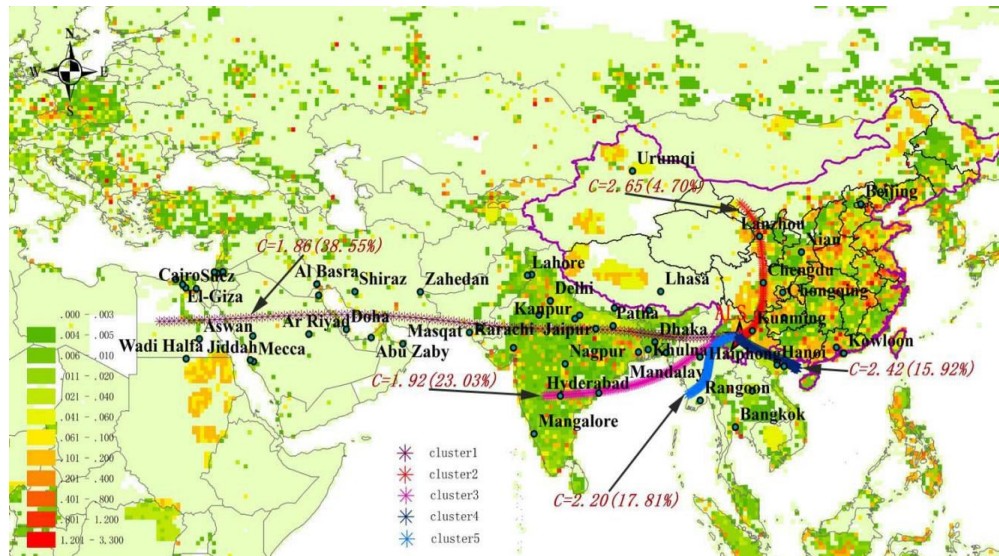




**Fig. 8:** The distribution of the Indian monsoon index (IMI), TGM and rainfall at ALS. IMI displays that the ISM
period was from May to September. RF events mainly appeared during the ISM period. Some high Hg events
were accompanied by NE and east wind events during the ISM period. High TGM events were accompanied by
the appearance of SW during the non-ISM period (October to April).

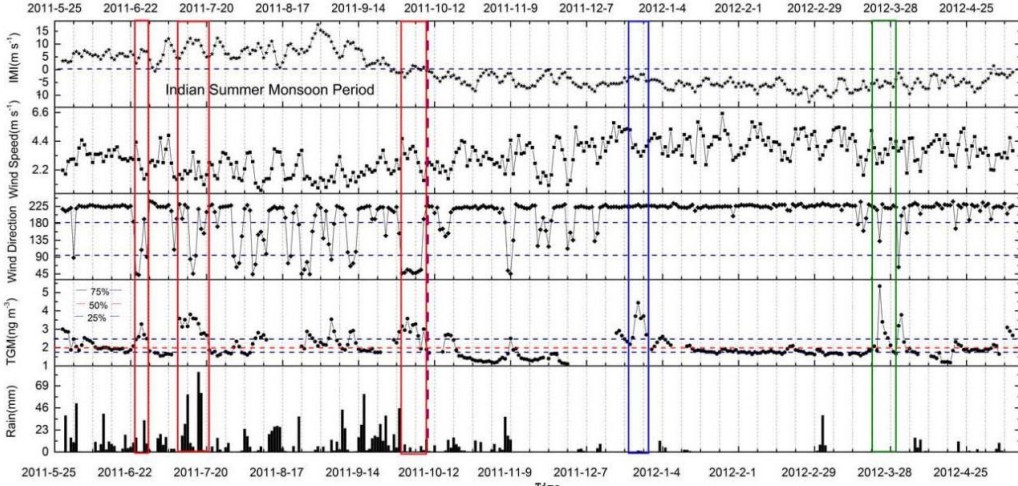






**Fig. 9:** Backward trajectories of air masses and the variation of TGM concentrations because of the strengthening
of and incursion of air flow from EASM on June 23-28, 2011 (a), and July 10-18, 2011 (b), during the ISM
period (May to September). The TGM level (C) were 3.27 ng m$^{-3}$(a) and 3.65 ng m$^{-3}$(b) with the air masses from
important industrial regions of inland China. The TGM level were down to 2.49 ng m$^{-3}$(a) and 3.02 ng m$^{-3}$(b)
while the air flow shifted to southwest and swept Southeast Asia.

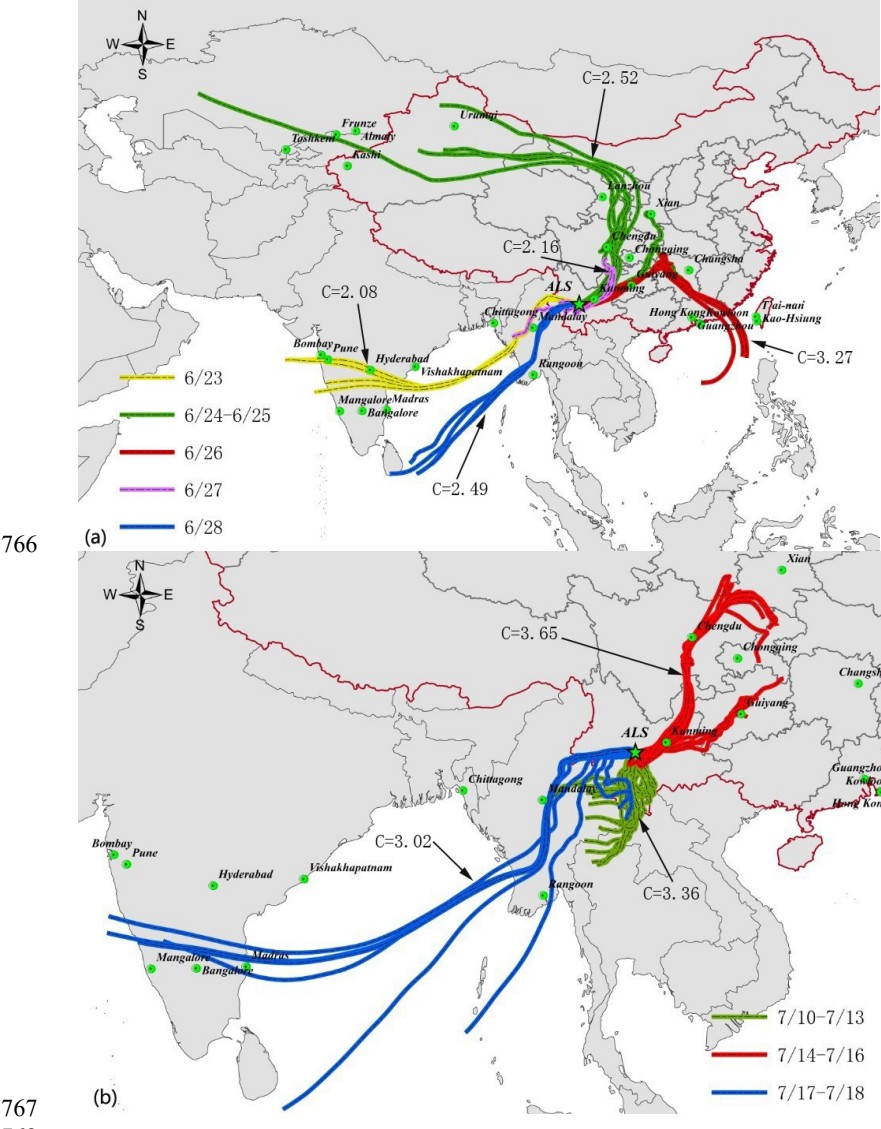

(a)
(b)





**Fig. 10:** Backward trajectories of air masses and the variation of TGM concentrations because of the
strengthening and incursion of air flow from EASM and the cold Siberian current from September 28 to October
9, 2011, during the non-ISM period (October to April). The TGM level (C) were 2.25 ng m⁻³, 3.18 ng m⁻³ and
2.53 ng m⁻³ with the air masses from inland China. The TGM level was down to 1.89 ng m⁻³ with the air masses
from South China Sea, northern Vietnam and Laos.

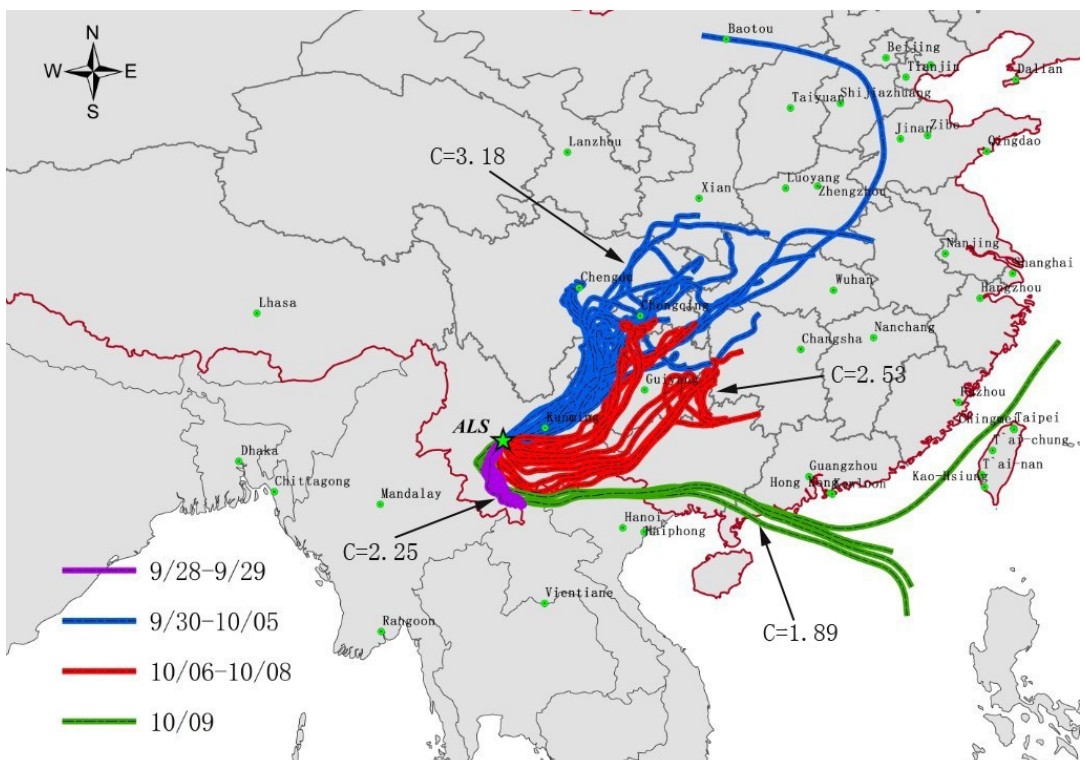





**Fig. 11:** Backward trajectories of air masses and the sites of fire events from December 23-31, 2011, during the
non-ISM period (October to April). The TGM level (C) was 3.13 ng m$^{-3}$ with air masses from important
Southeast Asia industrial regions. The TGM level dropped to 2.78 ng m$^{-3}$ with air masses from the Bay of Bengal.

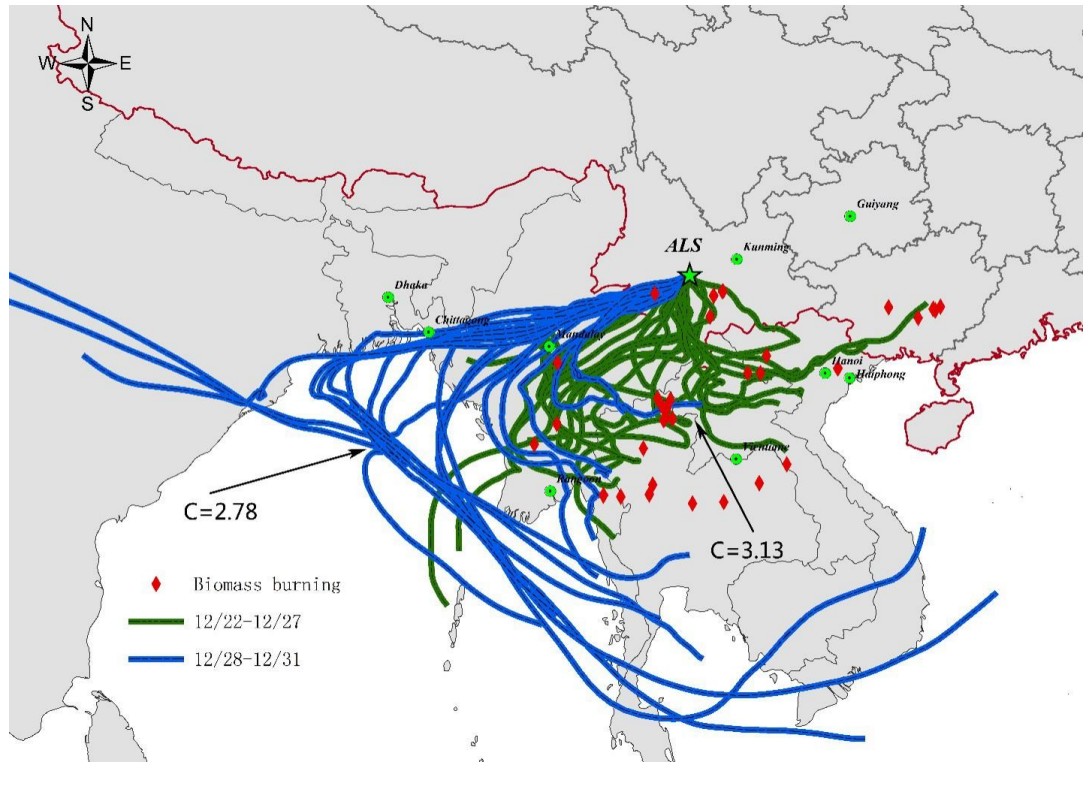






**Fig. 12:** Correlation of TGM and CO in December 23-31, 2011, during the non-ISM period (October to April).

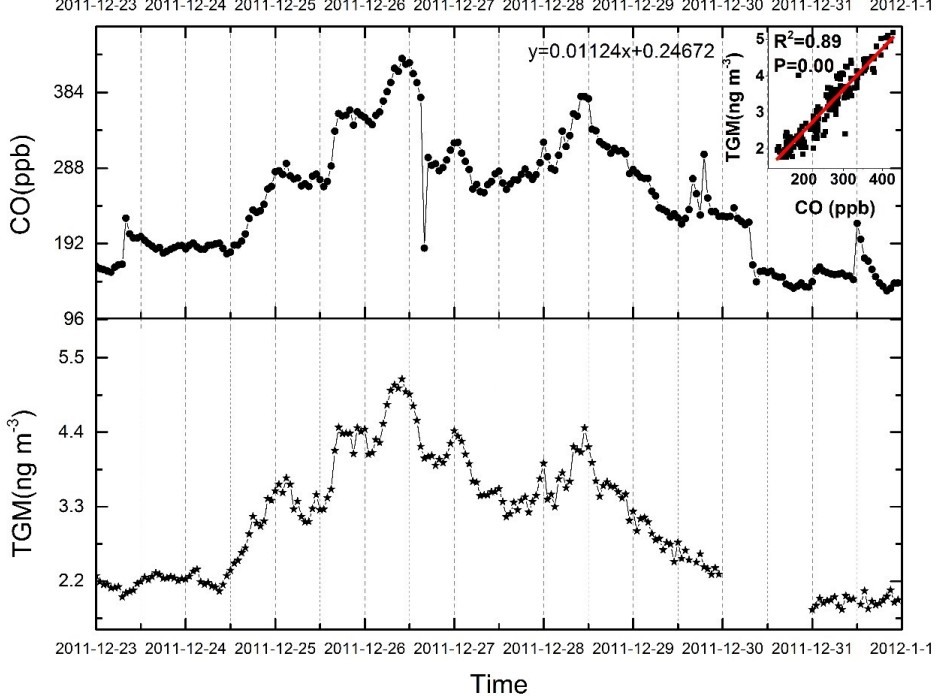







**Fig. 13:** Backward trajectories of air masses and the sites of fire events in March 23-29, 2012, during the non-
ISM period (October to April). The TGM level (C) were 4.53 ng m$^{-3}$ with the air masses from inland China. The
TGM level was 3.11 ng m$^{-3}$ with the air masses from Myanmar and high-frequency fire events.

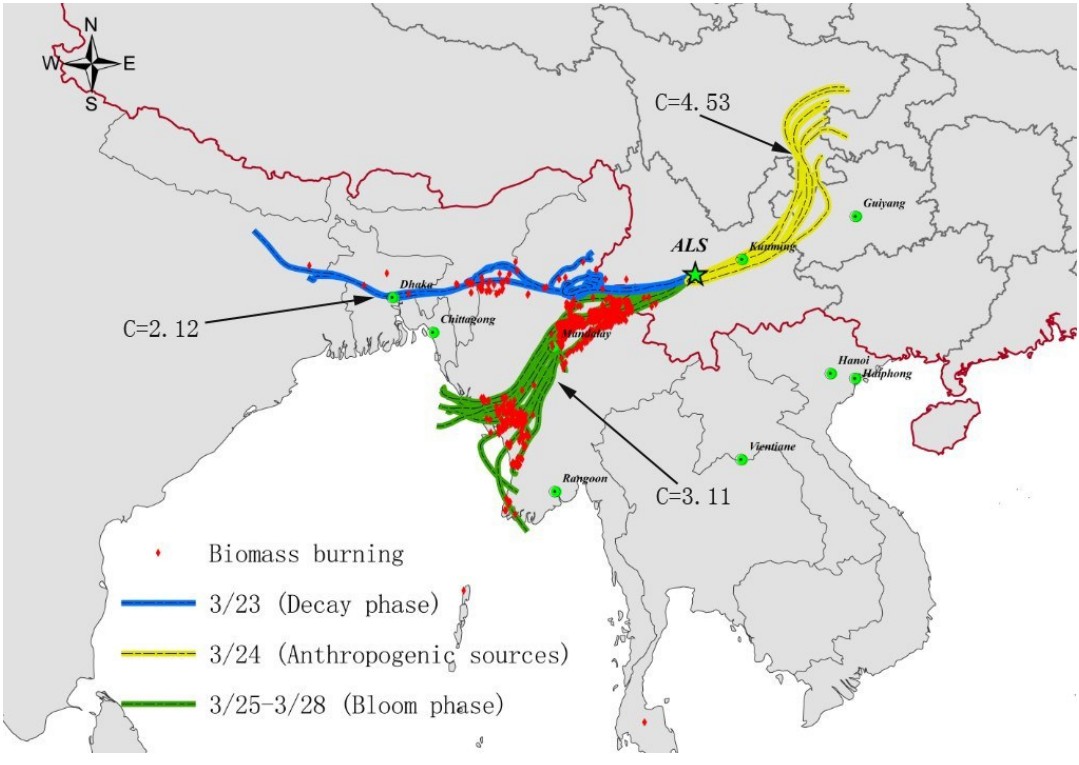




**Fig. 14:** Correlation of TGM and CO ($R^2$=0.53), correlations between TFRP (total fire radiative power) and CO
($R^2$=0.98) and TGM ($R^2$=0.45) in March 23-29, 2012, during the non-ISM period (October to April).

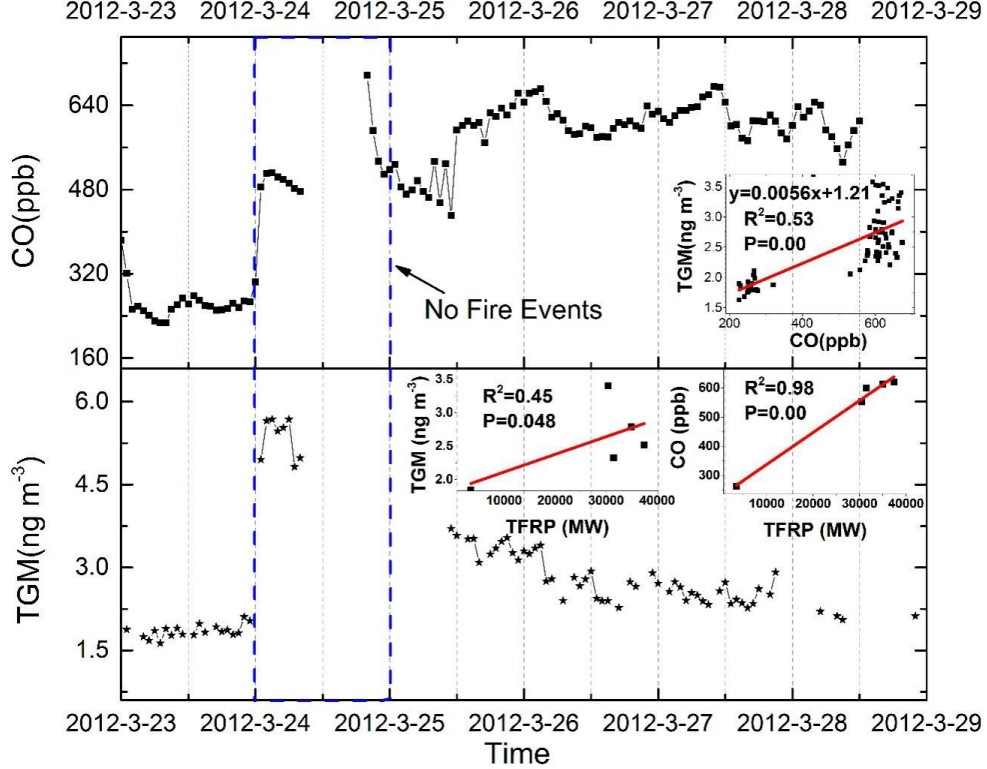







**Fig. 15:** Potential source regions and pathways of atmospheric TGM at ALS as identified by the CWT during
the ISM period (a, May to September) and the non-ISM period (b, October to April).

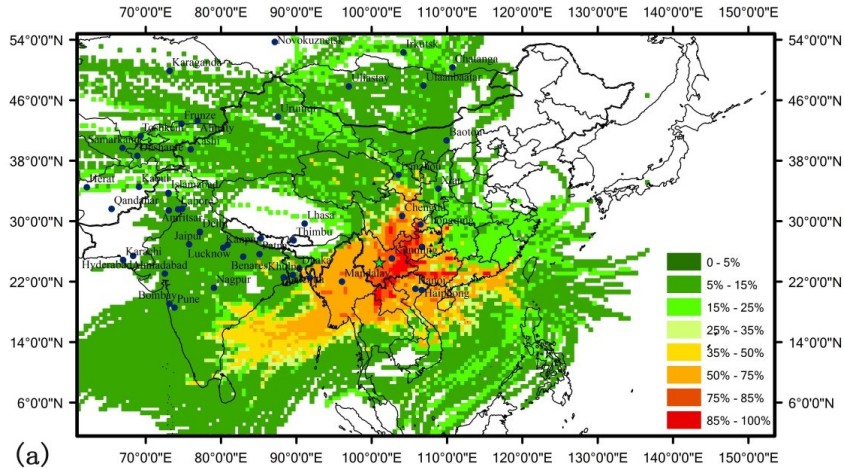


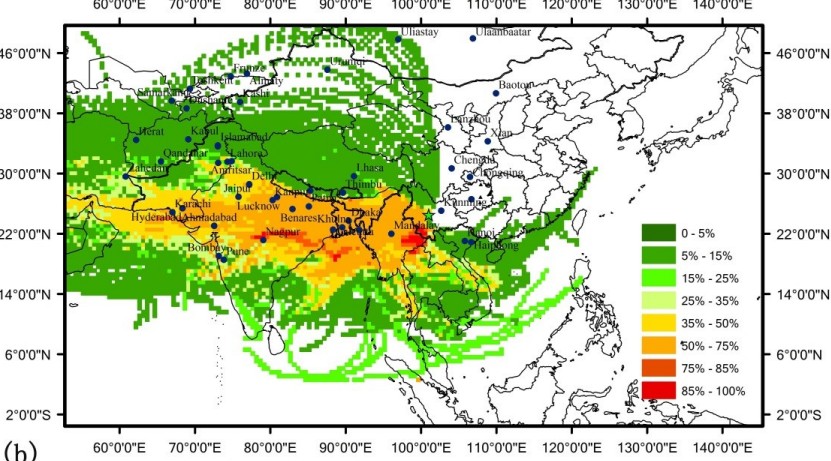
