# Peer review of "Monsoon-facilitated characteristics and transport of atmospheric mercury at a high-altitude background site in southwestern China"

_Atmospheric Chemistry and Physics, 2016_

## Referee Comment (RC1) · Anonymous Referee #1 · 3 Aug 2016

This paper presents TGM, GOM and PBM measurements over 1 year from a high altitude site in southwestern China. The authors have carried out a very thorough cluster and concentration weighted trajectory analysis of the data obtained and are able to attribute various findings to the influence of the seasonal monsoon. This is very nice work and these new measurements should be published, however the manuscript would benefit from a bit of restructuring and tidying up to make the conclusions clearer and so publishing should be subject to the following changes.

General comments 1. It needs to be clear that the monsoon is the driving force for the higher emissions rather than other seasonal factors such as lower oxidants, or higher source emissions. Please make sure the following questions are answered

clearly within the text. Are anthropogenic TGM events from inland China seen only in ISM/EASM conditions? Are biomass burning emissions (high CO) from South and Southeast Asia only seen in non-ISM or in all conditions?

2. Please be consistent when using TGM or Hg. Total Gaseous Mercury (TGM) was measured not Hg (e.g. Line 296 High Hg should be high TGM and Line 317 elevated Hg should be elevated TGM etc?) except during the GOM/PBM campaigns when Hg would have been measured instead. Any emission data would obviously be in terms of Hg.

Specific comments Abstract Towards the end of the abstract Lines 27-33 please separate the discussion of ISM, EASM and non-ISM, it comes across a bit confused (some more specific comments on the abstract are below).

Results and discussion The results section needs better headings, the monsoon discussion comes into all of the sections and so the sections are not as cleanly divided as first appears. I would suggest restructuring the results section starting with a discussion of the data compared to other sites and seasonal variation of the data, followed by the cluster analysis, CWT analysis and then a section on case study periods.

3.1. TGM, GOM, and PBM in ISM/EASM and non-ISM conditions. For this section merge the present Sections 3.1.1. and the seasonal discussion of GOM and PBM from Section 3.1.3. Move Figure 5 to Figure 3. When merging try not to repeat yourselves, in this section concentrate on discussing the statistics from Table 1 and the references to the old Figure 5 rather than on the interpretation of the data with respect to the monsoon. 3.2. Effect of the monsoon on the measurements (old Section 3.1.2). Include discussion on monthly TGM anomalies (old Figure 3), monsoon-facilitated effects, and finish with wind direction analysis (old Figure 4). 3.3 Cluster analysis (old Section 3.2) 3.4. Potential source regions of atmospheric Hg (old Section 3.4) rename Figure 15, Figure 8. 3.5 High TGM event case studies and the influence of monsoons, (old Section 3.3), use subheadings of dates to split case studies and reference to different figures?

Table 1 Could you possibly include CO statistics in the table?

Figures Figure 1: Can the position of ALS be made a little clearer? Figure 2: Can this be made clearer, widen plot to show the length of spikes? What is the time resolution of data? Figure 5: Are the GOM concentrations significantly higher in the ISM period? Figure 6: What sampling time do these points represent, hourly measurements or are they just from during the high Hg events? If these show all the data then these wind plots are not very convincing, most of the data comes from the SW but it is not clear from these that the higher events are all from the SW. The data looks better in the annotated trajectory plots, you could remove these and associated text in lines 265-274?

Technical corrections Lines 20-22, Quote some numbers to illustrate the higher concentrations in the ISM /EASM compared to the non-ISM period. Line 30 Consequently, southwestern........ sentence not needed as stated earlier in the abstract. Line 32 Change "should be" to "are thought to be" Line 40, remove "Therefore" and add instead "The monsoonal climate has the potential to strongly affect......" Line 57, remove "pose an" and just use "can impact other regions". Line 59, change "has" to "have". Line 61, remove "/regarding" Line 91, what height were you measuring at? Line 144 Are 500m trajectories valid? Line 92, Refer to Table 1 for seasonal breakdown of temperature and rainfall. Line 129, do you mean May to September and October to April for ISM and Non-ISM conditions respectively? Lines 165-171, Which comparison sites are located at a similar altitude to ALS? Are they all "background sites"? Line 160, What time resolution is plotted in Figure 2? Line 172 and Line 174, How long do the peaks last, is the length of time different for the ISM and non-ISM period, does this indicate different sources for each period? Line 175, "adjacent" not "adjacently" Line 233, remove "monsoonal" Lines 261 and 262, remove "wind" Line 267, "influencing" Line 344, remove "meanwhile"

Figure 7: Clusters defined here should then be referenced in subsequent figures. . ... 9, 10, 11, 13 relate to cluster analysis? Figures 11, 13, Where was the fire data from, please reference?

[Figure]

---

## Referee Comment (RC3) · Anonymous Referee #2 · 10 Aug 2016

A peer review of the manuscript acp-2016-506

Journal: ACP Title: Monsoon-facilitated characteristics and transport of atmospheric mercury at a high-altitude background site in southwestern China Author(s): H. Zhang et al. MS No.: acp-2016-506 MS Type: Research article Special Issue: Global Mercury Observation System – Atmosphere (GMOS-A)

General comments

The topic of the manuscript is important and pertinent to the special issue of the journal. The paper presents new data of long-term monitoring of air mercury and mercury speciation within GMOS network obtained at the Ailaoshan monitoring station, southwestern China. The title clearly reflects the contents of the paper, the main points of the research tasks, methodology of the data acquisition and calculation are described in the Abstracts. The measurements have been made using unified GMOS standard operational procedures for total mercury and mercury speciation. The emphasis is on the study of the reasons of the atmospheric mercury variations, dependence on Indian Summer (ISM) and East Asia Summer (EASM) Monsoons, long-term atmospheric transfer, and discussion on characters and geographical location of mercury emission sources. The paper is based on comprehensive mercury monitoring and meteorological data for backward trajectory calculation, which have provided sound evidence of mercury transfer with ISM and EASM air masses and explanation of the air mercury variations in southwestern China based on monitoring at the Ailaoshan station. The obtained data, calculations, and discussions are well structured and presented in the text and Conclusions. The manuscript has a comprehensive reference list.

Specific comments

1). Commonly authors discuss a long-distance transfer for all mercury species, e.g., at lines 272-274: "These westerlies could take the Hg from South Asia and Southeast Asia into southwestern China. Thus, the dependence of atmospheric Hg species on wind was likely attributed to an interplay of regional sources and the long-range transboundary transport of Hg". The drastic difference in the lifetime for air mercury species is well known. Gaseous elemental mercury (GEM, or Hg(0)) dominated in air has the longest lifetime in atmosphere (about one year) that provides really long-distance transfer for dozens thousand kilometers, whereas the lifetime of GOM is about 1 day, and it can be transported from its emission source to a distance only of 300-500 km. Thus, sources and origins of mercury species registered at the site can be different.

2). A reference to dependence of the GOM measurement on air humidity (lines 118-121 and 279-280): "Several previous studies reported that different GOM compounds (HgCl2, HgBr2 and HgO) have different collection efficiencies for the KCl-coated denuder surface, as high relative humidity can passivate KCl-coated denuder and make

GOM recoveries decrease (Huang et al., 2013a;Gustin et al., 2015;Huang and Gustin, 2015)". "A new study reported that high RH could reduce the collection of GOM by the KCl-coated denuder (Huang, Gustin et al. 2015). This could be another reason why the GOM was low in summer". In this respect, it is very important to compare possible range of uncertainty of the GOM measurement due to humidity variation, with the GOM measured values to confirm or cast doubt on real reason of the GOM variations.

3). It is not clear what parameters were measured at the Ailaoshan station along with mercury and meteoparameters, particularly, if the discussed CO concentration was measured at the station.

4). Authors mention SO2 measurement only once, in Introduction. The monitoring data of acid gases (such as SO2, NOX) are very useful for mercury emission source identification, e.g. for separating mercury emitting by forest fires or biomass burning from coal combustion plumes. That can be useful for future research.

Technical corrections

1). Various writing, compare: 66 . . . trans-boundary transport 68 . . . associated trans-boundary transport

2). A misprint at line 84: 82 . . .. to establish a global 83 mercury monitoring network for ambient concentrations and deposition of Hg though ground-based 84 observational platforms and oceanographic aircraft campaigns (Sprovieri et al., 2013)

Obviously, here should be 82 . . .. to establish a global 83 mercury monitoring network for ambient concentrations and deposition of Hg though ground-based 84 observational platforms, oceanographic, and aircraft campaigns (Sprovieri et al., 2013)

3).Lines 110-111 PBM ($\leq$0.2 $\mu$m) were removed using a 47 mm diameter Teflon filter (pore size 0.2 111 $\mu$m). It seems, the correct should be PBM ($\geq$0.2 $\mu$m) were removed, or PBM ($\leq$0.2 $\mu$m) were collected

Conclusion

[Figure]

The manuscript No acp-2016-506 can be accepted for Special Issue "Global Mercury Observation System – Atmosphere" (GMOS-A) with minor corrections.

---

## Author Comment (AC1) · 22 Sep 2016

AC: We thank the reviewer for the supportive comments regarding the merits of our work. We have carefully considered the reviewer's general and specific comments and revised the manuscript according to the reviewer's suggestions.

The RC1 is the same as the RC2 and the response to RC1 can be seen in the response to RC2.

---

## Author Comment (AC2) · 22 Sep 2016

RC- Reviewer's Comments; AC – Authors' Comments

RC1: This paper presents TGM, GOM and PBM measurements over 1 year from a high altitude site in southwestern China. The authors have carried out a very thorough cluster and concentration weighted trajectory analysis of the data obtained and are able to attribute various findings to the influence of the seasonal monsoon. This is very nice work and these new measurements should be published, however the manuscript would benefit from a bit of restructuring and tidying up to make the conclusions clearer and so publishing should be subject to the following changes.

[Figure]

AC: We thank the reviewer for the supportive comments regarding the merits of our work. We have carefully considered the reviewer's general and specific comments and revised the manuscript according to the reviewer's suggestions.

General comments

RC2: It needs to be clear that the monsoon is the driving force for the higher emissions rather than other seasonal factors such as lower oxidants, or higher source emissions. Please make sure the following questions are answered clearly within the text. Are anthropogenic TGM events from inland China seen only in ISM/EASM conditions? Are biomass burning emissions (high CO) from South and Southeast Asia only seen in non-ISM or in all conditions?

AC: TGM events from anthropogenic emission in inland China were both observed during ISM/EASM and non-ISM periods. However, the anthropogenic TGM events were mainly observed during ISM/EASM period, which were related to the long-range transport of atmospheric Hg from southwestern and southeastern China. Discussion regarding to these anthropogenic TGM events from inland China are given in on page 6, lines 187-201 of the revised manuscript. The concentration of CO was not measured in the ISM period of this study (CO was measured from October 2011 to May 2012). However, biomass burning events (high CO) were observed only during the non-ISM period and the high TGM events are associated with transport of air masses passing through know fire events (Wang et al., 2015). This discussion is provided on page 11, lines 368-390 of the revised manuscript.

RC3: Please be consistent when using TGM or Hg. Total Gaseous Mercury (TGM) was measured not Hg (e.g. Line 296 High Hg should be high TGM and Line 317 elevated Hg should be elevated TGM etc?) except during the GOM/PBM campaigns when Hg would have been measured instead. Any emission data would obviously be in terms of Hg.

AC: The term has been kept consistently throughout the revised manuscript.

Specific comments

RC4: Abstract. Towards the end of the abstract Lines 27-33 please separate the discussion of ISM, EASM and non-ISM, it comes across a bit confused (some more specific comments on the abstract are below).

AC: We introduced the characteristics of monsoon climate in East and South Asia in the Introduction section (lines 37-51, page 2). ISM and EASM occur simultaneously from May to September. Therefore, it is impossible to distinguish the EASM and ISM and the air flow to ALS is mainly controlled by the Indian monsoon climate, so we only separated the discussion of ISM and non-ISM periods. In this paper, the peaks of high TGM concentrations (Source 1 in Fig.2, page 23) during the ISM period (May to September) were caused by anthropogenic Hg emissions from inland China due to the strengthening of EASM.

Results and discussion

RC 5: The results section needs better headings, the monsoon discussion comes into all of the sections and so the sections are not as cleanly divided as first appears. I would suggest restructuring the results section starting with a discussion of the data compared to other sites and seasonal variation of the data, followed by the cluster analysis, CWT analysis and then a section on case study periods e.g.

AC: We have revised a significant portion of the text in Sections 3.1 and 3.2 to reflect the structure that the reviewer suggested. To better illustrate the potential sources of atmospheric Hg at ALS, we discussed the CWT results toward the end of Results & Discussion section in revised manuscript (lines 393-424, page 12).

RC 6: 3.1. TGM, GOM, and PBM in ISM/EASM and non-ISM conditions. For this section merge the present Sections 3.1.1. and the seasonal discussion of GOM and PBM from Section 3.1.3. Move Figure 5 to Figure 3. When merging try not to repeat yourselves, in this section concentrate on discussing the statistics from Table 1 and the

references to the old Figure 5 rather than on the interpretation of the data with respect to the monsoon.

AC: This is a good suggestion. We have changed the presentation structure of the revised manuscript as suggested by the reviewer. The distribution frequency of elevated TGM concentrations (> 2.09 ng m-3) under different wind directions (Fig. 4 in the original manuscript) has been moved to Section 3.1 in the revised manuscript (lines 187-201, page 6). The distribution of GOM and PBM has also been moved to section 3.1 (lines 203-218, pages 6&7). The seasonal statistics and analysis of daily averages for atmospheric Hg species with meteorological parameters have been moved to Section 3.2 (lines 221-227, page 7). Monsoonal distribution patterns of TGM, GOM and PBM based on the four sampling campaigns in ALS (Fig. 5 in the original manuscript) has also been moved (pages 25 & 26). The discussion of TGM, GOM and PBM distribution in revised manuscript has been made clear after the text re-structuring.

RC 7: 3.2. Effect of the monsoon on the measurements (old Section 3.1.2). Include discussion on monthly TGM anomalies (old Figure 3), monsoon-facilitated effects, and finish with wind direction analysis (old Figure 4).

AC: In the revised manuscript, we deleted the old section 3.1.2 and discussed the monthly variations of TGM (new Fig.4, lines 229-237, page 7), monsoon-facilitated effects (new Fig. 5, lines 258-266, page 8), and finish with wind direction analysis (Fig. 6, lines 268-277, page 8).

RC 8: 3.3 Cluster analysis (old Section 3.2)

AC: We revised the title of Section 3.2 to "Transboundary transport of Hg facilitated by monsoons" after re-structuring the manuscript (new section 3.3, lines 290-324).

RC 9: 3.4. Potential source regions of atmospheric Hg (old Section 3.4) rename Figure 15, Figure 8.

AC: We have revised the manuscript structure (new Section 3.5) and the Title of Figure

15.

RC 10: 3.5. High TGM event case studies and the influence of monsoons, (old Section 3.3), use subheadings of dates to split case studies and reference to different figures.

AC: We put the high TGM events case studies and the influence of monsoon in Section 3.4 of the revised manuscript (lines 326-390, page 10-11). The subheading of the dates of the events were shown in Fig.9, Fig.10, Fig 11 and Fig.13, respectively, in the revised manuscript.

RC11: Table 1, Could you possibly include CO statistics in the table?

AC: CO concentrations were only measured during the non-ISM period (from October 2011 to May 2012) in this study. The statistics of the CO concentrations have been added in Table 1 in the revised manuscript (Page 21).

Figures

RC 12: Figure 1: Can the position of ALS be made a little clearer?

AC: Yes, Figure 1 was revised in the revised manuscript to show the location of ALS

RC 13: Figure 2: Can this be made clearer, widen plot to show the length of spikes? What is the time resolution of data?

AC: The time resolution Figure 2 is five minutes. We have modified Figure 2 to show these spikes clearly in the revised manuscript.

RC 14: Figure 5: Are the GOM concentrations significantly higher in the ISM period?

AC: Yes, the mean GOM concentration (2.45 pg m-3) during the ISM period is higher than that (2.06 pg m-3) during the non-ISM period. The statistics of mean GOM concentrations during ISM/EASM and non-ISM periods have been shown in Table 1 in the revised manuscript (page 21).

RC 15: Figure 6: What sampling time do these points represent, hourly measurements

or are they just from during the high Hg events? If these show all the data then these wind plots are not very convincing, most of the data comes from the SW but it is not clear from these that the higher events are all from the SW. The data looks better in the annotated trajectory plots, you could remove these and associated text in lines 265-274?

AC: The sampling time in Figure 6 represents the periods of the PBM and GOM measurements: August 17–24, 2011, December 3–17, 2011, April 12–19, 2012, and July 11–21, 2012 (lines 129-131, page 4). We used the hourly TGM, GOM and PBM to make the plots, these wind-dot plots show the distribution of TGM, GOM and PBM along with the wind direction, and indicate that SW and NE winds are associated with the high Hg events during the ISM period. Nearly all TGM, GOM and PBM events occurred during the non-ISM period due to strong westerlies.

Technical corrections

We thank the reviewer taking the time for these editorial recommendations.

RC 16: Lines 20-22, Quote some numbers to illustrate the higher concentrations in the ISM /EASM compared to the non-ISM period.

AC: We have included the numbers as suggested in the revised manuscript (lines 21-22, page 1).

RC 17: Line 30 Consequently, southwestern: : :: : :.. sentence not needed as stated earlier in the abstract.

AC: We have deleted the sentence as suggested in the revised manuscript (line 30, page 1).

RC 18: Line 32 Change "should be" to "are thought to be"

AC: We have revised the sentence as suggested in the revised manuscript (line 31, page 1).

RC 19: Line 40, remove "Therefore" and add instead "The monsoonal climate has the potential to strongly affect: : :: : :" AC: We have revised the sentence as suggested in the revised manuscript (line 40, page 2).

RC 20: Line 57, remove "pose an" and just use "can impact other regions". AC: We have revised the sentence as suggested in the revised manuscript (line 57-58, page 2).

RC 21: Line 59, change "has" to "have".

AC: We have revised the wording as suggested in the revised manuscript (line 60, page 2).

RC 22: Line 61, remove "/regarding"

AC: We have revised the wording as suggested in the revised manuscript (line 62, page 2).

RC 23: Line 91, what height were you measuring at? Line 144 Are 500m trajectories valid? AC: We thank the reviewer for pointing this out, the altitude of the monitoring site is 2500 m. An altitude of at 500 m above was used, which is the height typically in the mixing layer.

RC 24: Line 92, Refer to Table 1 for seasonal breakdown of temperature and rainfall.

AC: We have referred to Table 1 as suggested (lines 93-94, page 3).

RC 25: Line 129, do you mean May to September and October to April for ISM and Non-ISM conditions respectively?

AC: We thank the reviewer for pointing this out and have revised the wording in the revised manuscript (lines 130-131, page 4).

RC 26: Lines 165-171, Which comparison sites are located at a similar altitude to ALS? Are they all "background sites"?

[Figure]

AC: Yes, these sites are background sites (Fu et al., 2015). The altitude (2178 m) of Mt. Leigong in Guizhou province is similar to that of ALS. The altitude of Shangri-La Baseline Observatory and Mt. Waliguan (WLG) Baseline Observatory in Tibetan plateau are higher than 3500 m. The altitude of Mt. Changbai and Mt. Gongga are ∼2000 m.

RC 276: Line 160, What time resolution is plotted in Figure 2?

AC: The time resolution of TGM is five minutes in Figure 2. We have added the caption in Figure 2 (Page 23).

RC 28: Line 172 and Line 174, How long do the peaks last, is the length of time different for the ISM and non-ISM period, does this indicate different sources for each period?

AC: These peaks lasted for 2-6 days, and the length of peak time during the non-ISM period was longer than that during the ISM period (Figures 9, 10, 11, 13), indicating that the biomass burning from Southeast Asia could last for a long time during non-ISM period.

RC 29: Line 175, "adjacent" not "adjacently"

AC: We have revised the wording in the revised manuscript (line 177, page 5).

RC 30: Line 233, remove "monsoonal"

AC: We have removed "monsoonal" in the revised manuscript (line 223, page 7).

RC 31: Lines 261 and 262, remove "wind" AC: We have removed "wind" in the revised manuscript (line 264-265, page 8).

RC 32: Line 267, "influencing" AC: We have revised the wording in the revised manuscript (line 270, page 8).

RC 33: Line 344, remove "meanwhile"

[Figure]

AC: We have removed "meanwhile" in the revised manuscript (line 348, page 10).

RC 34: Figure 7: Clusters defined here should then be referenced in subsequent figures: : :. 9, 10, 11, 13 relate to cluster analysis?

AC: We appreciate the reviewer's insightful comment. In the structure of our paper, the discussion of the data compared to other sites and seasonal variation of the data need to be merged into the cluster analysis about transboundary transport of Hg facilitated by monsoons, then followed by a section on case study periods. It is convenient to explain the high TGM peaks were from different sources for ISM and non-ISM period in the section on case study periods.

RC 35: Figures 11, 13, Where was the fire data from, please reference?

AC: We thank the reviewer for pointing this out and have added the location of fire data in Figures 11, 13 in the revised manuscript. The fire data are from NASA's satellite products at https://firms.modaps.eosdis.nasa.gov/firemap/ (line 787, page 32; line 799, page 34).

---

## Author Comment (AC3) · 22 Sep 2016

General comments

RC 1: The topic of the manuscript is important and pertinent to the special issue of the journal.

The paper presents new data of long-term monitoring of air mercury and mercury speciation within GMOS network obtained at the Ailaoshan monitoring station, southwestern China. The title clearly reflects the contents of the paper, the main points of the research tasks, methodology of the data acquisition and calculation are described in the Abstracts. The measurements have been made using unified GMOS standard

operational procedures for total mercury and mercury speciation. The emphasis is on the study of the reasons of the atmospheric mercury variations, dependence on Indian Summer (ISM) and East Asia Summer (EASM) Monsoons, long-term atmospheric transfer, and discussion on characters and geographical location of mercury emission sources. The paper is based on comprehensive mercury monitoring and meteorological data for backward trajectory calculation, which have provided sound evidence of mercury transfer with ISM and EASM air masses and explanation of the air mercury variations in southwestern China based on monitoring at the Ailaoshan station. The obtained data, calculations, and discussions are well structured and presented in the text and Conclusions. The manuscript has a comprehensive reference list. AC: We appreciate the reviewer's insights and supportive comments for our study. Despite the difficult physical conditions at ALS, we were able to collect the data with the support of the GMOS program. We very much appreciate the vision of GMOS and it is our goal to share the data with Hg research community to advance the understanding of atmospheric Hg transport in Asian region under the influence of monsoonal weather.

Specific comments

RC 2: 1). Commonly authors discuss a long-distance transfer for all mercury species, e.g., at lines 272-274: "These westerlies could take the Hg from South Asia and Southeast Asia into southwestern China. Thus, the dependence of atmospheric Hg species on wind was likely attributed to an interplay of regional sources and the long-range transboundary transport of Hg". The drastic difference in the lifetime for air mercury species is well known. Gaseous elemental mercury (GEM, or Hg(0)) dominated in air has the longest lifetime in atmosphere (about one year) that provides really long-distance transfer for dozens thousand kilometers, whereas the lifetime of GOM is about 1 day, and it can be transported from its emission source to a distance only of 300-500 km. Thus, sources and origins of mercury species registered at the site can be different.

AC: We thank the reviewer for pointing this out and have revised the statement as

suggested in the revised manuscript (lines 275-277, page 8).

RC 3: 2). A reference to dependence of the GOM measurement on air humidity (lines 118- 121 and 279-280): "Several previous studies reported that different GOM compounds (HgCl2, HgBr2 and HgO) have different collection efficiencies for the KCl-coated denuder surface, as high relative humidity can passivate KCl-coated denuder and make GOM recoveries decrease (Huang et al., 2013a;Gustin et al., 2015;Huang and Gustin, 2015)". "A new study reported that high RH could reduce the collection of GOM by the KCl-coated denuder (Huang, Gustin et al. 2015). This could be another reason why the GOM was low in summer". In this respect, it is very important to compare possible range of uncertainty of the GOM measurement due to humidity variation, with the GOM measured values to confirm or cast doubt on real reason of the GOM variations.

AC: We agree with the reviewer that the variation of humidity could affect the collection efficiencies of the KCl-coated denuder based on the findings of Gustin's group, and we also observed the extremely low GOM concentrations due to the high humidity at ALS during ISM period.

RC 4: 3). It is not clear what parameters were measured at the Ailaoshan station along with mercury and meteoparameters, particularly, if the discussed CO concentration was measured at the station.

AC: We just measured CO at the ALS along with mercury and meteoparameters, but due to the limited conditions at ALS, we just beginning to continuously measure CO from October 2011 and collected CO data in non-ISM period (The distribution of the Indian monsoon index (IMI), Wind direction, CO and TGM at ALS is shown in Figure 1).

RC 5: Authors mention SO2 measurement only once, in Introduction. The monitoring data of acid gases (such as SO2, NOX) are very useful for mercury emission source identification, e.g. for separating mercury emitting by forest fires or biomass burning

from coal combustion plumes. That can be useful for future research.

AC: Considered that the measurement of CO is sufficient to identify the potential sources, we appreciate the reviewer's insightful comment. We agree that criterion air pollutants such as CO, SO2, NOX and O3 are important for the data analysis of TGM.

Technical corrections

RC 6: 1). Various writing, compare: 66 : : : trans-boundary transport 68 : : : associated transboundary transport

AC: The wording has been revised as suggested in the revised manuscript (line 67, page 2).

RC7: 2). A misprint at line 84: 82 : : :. to establish a global 83 mercury monitoring network for ambient concentrations and deposition of Hg though ground-based 84 observational platforms and oceanographic aircraft campaigns (Sprovieri et al., 2013). Obviously, here should be 82 : : :. to establish a global 83 mercury monitoring network for ambient concentrations and deposition of Hg though ground-based 84 observational platforms, oceanographic, and aircraft campaigns (Sprovieri et al., 2013)

AC: The statement has been revised as suggested in the revised manuscript (lines 83-85, page 3).

RC 8: 3).Lines 110-111 PBM ($\leq$0.2 $\mu$m) were removed using a 47 mm diameter Teflon filter (pore size 0.2 $\mu$m). It seems, the correct should be PBM ($\geq$0.2 $\mu$m) were removed, or PBM ($\leq$0.2 $\mu$m) were collected

AC: We thank the reviewer for pointing this out and the statement has been revised as suggested in the revised manuscript (line 111, page 4). Reference: Wang, X., Zhang, H., Lin, C. J., Fu, X., Zhang, Y., and Feng, X.: Transboundary transport and deposition of Hg emission from springtime biomass burning in the Indo‐China Peninsula. Journal of Geophysical Research: Atmospheres, 120(18), 9758-9771, 2015. Fu, X. W., Zhang, H., Yu, B., Wang, X., Lin, C. J., and Feng, X. B.: Observations of atmospheric

mercury in China: a critical review, Atmos Chem Phys, 15, 9455-9476, 2015.

[Figure]

[Figure]

[Figure]

**Fig. 1.** The distribution of the Indian monsoon index (IMI), Wind direction, CO and TGM at ALS